# A native chemical chaperone in the human eye lens

**Eugene Serebryany[1], Sourav Chowdhury[1], Christopher N Woods[2], David C Thorn[1], Nicki E Watson[3], Arthur A McClelland[3], Rachel E Klevit[2], Eugene I Shakhnovich[1]\***

[1]Department of Chemistry and Chemical Biology, Harvard University, Cambridge, United States; [2]Department of Biochemistry, University of Washington, Seattle, United States; [3]Center for Nanoscale Systems, Harvard University, Cambridge, United States

**Abstract** Cataract is one of the most prevalent protein aggregation disorders and still the most common cause of vision loss worldwide. The metabolically quiescent core region of the human lens lacks cellular or protein turnover; it has therefore evolved remarkable mechanisms to resist light-scattering protein aggregation for a lifetime. We now report that one such mechanism involves an unusually abundant lens metabolite, *myo*-inositol, suppressing aggregation of lens crystallins. We quantified aggregation suppression using our previously well-characterized *in vitro* aggregation assays of oxidation-mimicking human γD-crystallin variants and investigated *myo*-inositol's molecular mechanism of action using solution NMR, negative-stain TEM, differential scanning fluorometry, thermal scanning Raman spectroscopy, turbidimetry in redox buffers, and free thiol quantitation. Unlike many known chemical chaperones, *myo*-inositol's primary target was not the native, unfolded, or final aggregated states of the protein; rather, we propose that it was the rate-limiting bimolecular step on the aggregation pathway. Given recent metabolomic evidence that it is severely depleted in human cataractous lenses compared to age-matched controls, we suggest that maintaining or restoring healthy levels of *myo*-inositol in the lens may be a simple, safe, and globally accessible strategy to prevent or delay lens opacification due to age-onset cataract.

## Editor's evaluation

Cataract is one of the most prevalent protein aggregation disorders and the leading cause of vision loss worldwide. As the eye lens lacks cellular or protein turnover, efficient mechanisms have evolved to prevent protein aggregation for a lifetime. This study shows that an abundant metabolite of the eye lens, myo-inositol, plays an important role in preventing aggregation, acting as a chemical chaperone by an unusual mechanism.

## Introduction

Cataract disease afflicts tens of millions of people per year and is the worldwide leading cause of blindness (*Lam et al., 2015*). The disease is caused by light-scattering aggregation of the extremely long-lived crystallin proteins in the lens (*Bloemendal et al., 2004*). This aggregation is associated with accumulation of post-translational modifications, including disulfide formation, UV- or metal-catalyzed Trp oxidation, deamidation, Asp isomerization, Lys derivatization, truncation, and various others (*Hains and Truscott, 2007*; *Serebryany and King, 2014*; *Su et al., 2011*). Most age-onset cataract forms in the oldest, nuclear (or core) region of the lens (*Bloemendal et al., 2004*). The cells in this region lack all organelles and thus protein synthesis and degradation capacity, and lens crystallins comprise ~90% of their protein mass (*Wride, 2011*). The most thermodynamically stable crystallin in

*For correspondence:
shakhnovich@chemistry.harvard.edu

**Competing interest:** The authors declare that no competing interests exist.

humans is γD-crystallin (HγD), which, at neutral pH, has a $T_m$ above 80 °C (*Serebryany et al., 2018*), resistance to 8 M urea (*Kosinski-Collins and King, 2003*), and unfolding half-life estimated to be on the order of decades (*Mills-Henry et al., 2019*). It is particularly abundant in the nuclear region of the lens and enriched in insoluble aggregates (*Su et al., 2011*). HγD consists of two homologous intercalated double-Greek key domains. The N-terminal domain is less stable and derives part of its stability from the domain interface (*Mills-Henry et al., 2019*; *Serebryany and King, 2014*). All γ-crystallins are Cys-rich (*Serebryany et al., 2021*); thus, HγD contains six Cys residues, four of which are buried in the core of the N-terminal domain, yet there are no disulfides in the native state of the protein *in vitro* or in the young lens (*Basak et al., 2003*; *Fan et al., 2015*; *Ramkumar et al., 2018*; *Serebryany et al., 2018*).

In prior research, we have developed a physiologically relevant *in vitro* model of cataract-associated aggregation and investigated in molecular mechanistic detail the unfolding pathway and probable interactions involved (*Serebryany and King, 2015*; *Serebryany et al., 2016a*; *Serebryany et al., 2016b*; *Serebryany et al., 2018*). Mutations in HγD that cause congenital cataracts, including W42R (*Wang et al., 2011*), cluster near the N-terminal β-hairpin (*Serebryany and King, 2014*). The vast majority of cataracts arise not from congenital mutations but from age-related changes in the lens, including Trp oxidation in lens crystallins (*Hains and Truscott, 2007*). We have shown that the oxidation-mimicking W42Q variant and the congenital W42R produce similarly strong aggregation under physiologically relevant oxidizing conditions (*Serebryany et al., 2016b*). The aggregates generated by the W42Q variant, or by a variant at the cognate site in the C-terminal domain (W130E), have been extensively characterized, making them well-suited for the present study (*Serebryany and King, 2015*; *Serebryany et al., 2016a*; *Serebryany et al., 2016b*). These variants are well-folded at pH 7 and 37 °C, yet rare partially misfolded intermediates formed under those conditions, if locked by an internal disulfide bond, proceed to form aggregates. Since aggregation crucially depends on non-native intramolecular disulfides, and due to latent oxidoreductase activity in HγD and likely other crystallins, this conformational transition from the native to the aggregation precursor state depends on the redox state of the crystallin proteome (*Serebryany et al., 2018*). These neutral-pH HγD aggregates do not stain with Thioflavin T, indicating that they are not amyloid fibrils (*Serebryany and King, 2015*; *Serebryany et al., 2016a*). This is consistent with electron microscopy of cataractous lens cytoplasm, which did not reveal any extended fibrillar structures (*Costello et al., 2012*; *Metlapally et al., 2008*). Others have also shown that crystallin aggregates induced by directly irradiating WT γ-crystallins with UV light are mostly (though not entirely) non-amyloid (*Moran et al., 2013*; *Roskamp et al., 2017*; *Schafheimer and King, 2013*). The aggregates' binding to the surface hydrophobicity probe bis-ANS is also limited and occurs mostly at the earliest stages of aggregation (*Serebryany and King, 2015*; *Serebryany et al., 2016a*). Therefore, the most experimentally tractable way of measuring the aggregation process is to measure light-scattering (solution turbidity) itself. This is also the most physiologically relevant metric since turbidity of the lens fiber cell cytoplasm is the main symptom of cataract. Fortunately, we have shown that solution turbidity is proportional to total mass for these aggregates (*Serebryany and King, 2015*) and that simple power laws relate the aggregation rate and lag time to protein concentration, such that all aggregation traces collapse onto a single master curve, where all points that map onto a given master-curve point have the same particle size distribution (*Serebryany et al., 2019*).

Despite intense research interest (*Makley et al., 2015*; *Shanmugam et al., 2015*; *Zhao et al., 2015*), no preventative or therapeutic drugs against cataract have been approved to-date, leaving surgery as the only treatment option. Aggregate costs of cataract surgery are high in high-income countries (*Frick et al., 2007*), while availability and outcomes are often poor in low- and middle-income ones (*Kara-Junior et al., 2017*; *Lam et al., 2015*). Hence, most cataract around the world remains untreated (*WHO, 2014*). Delaying the age of onset of the disease by ~14% would cut the need for surgery by half by pushing the onset of blindness beyond the average lifespan (*WHO, 2014*).

However, the search for cataract treatments has been hampered by three major challenges. Any such treatment must be (1) able to permeate into the nuclear region of the lens where most cataract occurs; (2) simple to use (e.g. eye drops) and extremely safe; and (3) sufficiently stable and cheap to be available to the vast majority of cataract patients who do not reside in high-income countries (*Khairallah et al., 2015*; *Wang et al., 2016*). The lens is a tightly packed tissue, and a diffusion barrier that forms during middle age prevents almost all externally applied compounds from penetrating into the

lens nucleus (*Heikkinen et al., 2019*). Unless a drug candidate lies within the highly constrained space of lens-permeating compounds, it is unlikely to be globally useful.

We therefore wondered whether the small molecule metabolome of the lens cytoplasm itself has evolved to produce or concentrate metabolites that suppress crystallin aggregation. Strong evolutionary pressure to preserve clear vision has produced multiple mechanisms to delay crystallin aggregation in the lens for as long as possible. Prior research focused largely on the crystallins themselves: the thermodynamic stability, kinetic stability, and redox activity of monomeric γ-crystallins (*Roskamp et al., 2020*; *Serebryany and King, 2014*) the stabilizing oligomerization and destabilizing deamidation of β-crystallins (*Lampi et al., 2014*) and the passive chaperone role of α-crystallins (*Slingsby et al., 2013*). Aggregation of crystallins, like that of many other proteins, can be modulated by osmolytes or other small metabolites (*Attanasio et al., 2007*; *Goulet et al., 2011*). Metabolomic analyses showed certain small metabolites to be unusually abundant in the lens, notably *myo*-inositol (*Tomana et al., 1984*; *Tsentalovich et al., 2015*) – which, however, is greatly depleted in cataractous lenses (*Tsentalovich et al., 2015*; *Yanshole et al., 2019*). We now report that *myo*-inositol (commonly referred to as just 'inositol', since it is the naturally prevalent isomer) acts as a kinetic inhibitor of the aggregation of the oxidation-mimicking, cataract-associated W42Q variant of human γD-crystallin, as well as other variants that aggregate under physiologically relevant oxidizing conditions. Notably, this compound slows the aggregation rate even when added at a later stage in the aggregation process. We present evidence that inositol inhibits the rate-limiting bimolecular step in the aggregation pathway and thereby indirectly disfavors the non-native disulfide bonds that stabilize the aggregation precursor. As one of the very few molecules known to permeate into the lens from the outside, by both passive and active transport (*Cotlier, 1970*; *Diecke et al., 1995*), *myo*-inositol therefore has the potential to become a safe, simple, and globally accessible treatment to prevent or delay loss of vision due to age-onset cataract disease.

## Results

### Inositol suppresses aggregation of cataract-associated γD-crystallin variants

We tested a variety of sugars and sugar alcohols for their ability to suppress aggregation of W42Q HγD as the *in vitro* model of cataract-associated aggregation. *Figure 1A* shows a representative set of normalized turbidity traces demonstrating strong and dose-dependent suppression of turbidity development by *myo*-inositol. As shown in *Figure 1B*, even highly similar compounds, at 100 mM each, had widely varying effects, from strong suppression to moderate enhancement of aggregation. Solution turbidity at the end of the 4.5 hr incubation was measured in 96-well plates and the turbid solutions then clarified by centrifugation. Consistent with prior studies (see above), the amount of residual protein in the clarified supernatant had a strong linear inverse correlation with end-point solution turbidity, suggesting that these compounds inhibited the aggregation process without changing aggregate shape or size. The linear correlation between turbidity and aggregate mass further validated the use of turbidity as a quantitative measure of this particular aggregation process. The concentrations of compounds used here were not high enough to alter bulk solution properties (*Figure 1—figure supplement 1*).

That the ability to suppress crystallin aggregation varied among chemically similar compounds strongly suggests specific protein-metabolite interactions. Thus, *myo*-inositol was much more effective than mannitol – a sugar alcohol with the same number of carbons and hydroxyls but a linear rather than cyclic structure. Galactose was one of the strongest aggregation suppressors, yet its derivative IPTG enhanced aggregation.

*Figure 1C* shows a more detailed comparison of dose-response curves for *myo*-inositol and glycerol. Tsentalovich et al. reported 24.6±6.7 μmol of *myo*-inositol per gram of wet lens in the nuclear region, dropping to just 1.9±1.8 μmol/g in the nuclear region of cataractous lenses (*Tsentalovich et al., 2015*). Assuming a water content of ~60% and a total of ~50% of the water in a bound state, per a prior study (*Heys et al., 2008*), we arrive at an estimate of 82±22 mM *myo*-inositol in the free water fraction of the healthy human lens. In this concentration range, *myo*-inositol already had a substantial effect in our assay, suppressing the rate of turbidity development by ~35% and increasing the lag time by a similar amount (*Figure 1D*), consistent with the 'master curve' relationships among

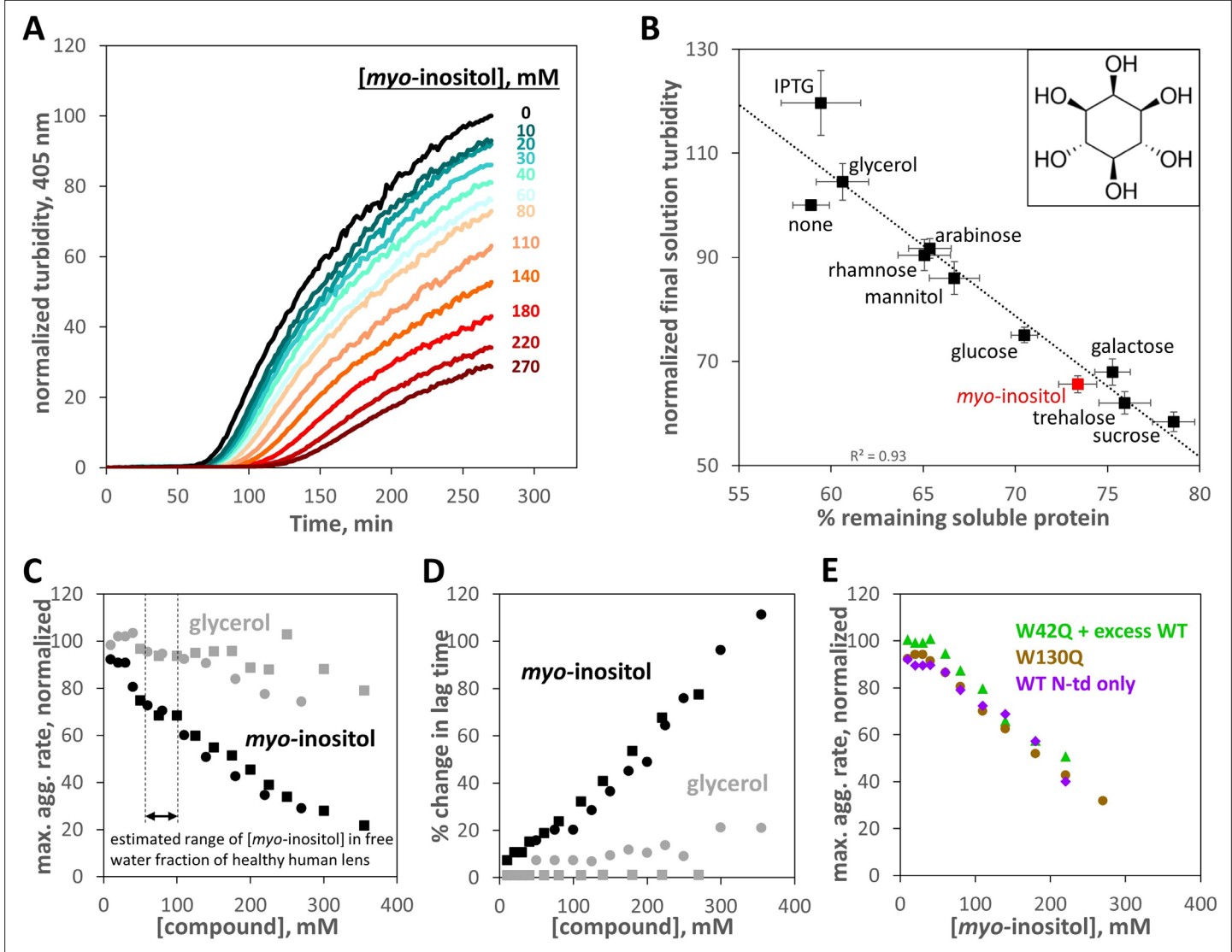

**Figure 1.** Suppression of human γD-crystallin aggregation by myo-inositol. Oxidative aggregation of the cataract-mimicking W42Q variant of HγD was initiated as previously described (*Serebryany et al., 2018*). (**A**) Normalized turbidity traces for the oxidative aggregation of 40 μM HγD W42Q with varying concentrations of *myo*-inositol. (**B**) Sugars and sugar alcohols, each at 100 mM, suppressed HγD W42Q aggregation to varying degrees. Isopropyl-β-D-thiogalactoside (IPTG) moderately enhanced turbidity development. *Myo*-inositol (structure shown in inset) consistently and strongly suppressed turbidity development, second only to the disaccharides, trehalose and sucrose. A total of 8 independent replicates, at pH 7.4 (HEPES), pH 6.7 (PIPES), or pH 6.0 (MES) and 150 mM NaCl all produced similar results and were averaged together. Notably, the strong linear correlation between reduction in solution turbidity and increase in the proportion of protein remaining soluble indicated these compounds (except perhaps IPTG) did not significantly change aggregate geometry. Raw data are available as *Figure 1—source data 1*. (**C**) Dose response for aggregation suppression by *myo*-inositol compared to glycerol. Notably, *myo*-inositol had a significant effect in the physiological concentration range. Data from two independent replicates (*circles, squares*) are shown; *black circles* correspond to the data in panel A. All aggregation rates were normalized to the rate without *myo*-inositol. (**D**) Percent change in aggregation lag time for the same experiments as in Panel C. (**E**) Suppression of oxidative aggregation by *myo*-inositol generalizes to other HγD constructs: *Green triangles*, 20 μM W42Q whose aggregation is catalyzed by 180 μM WT HγD at 37 °C; *Beige circles*, 40 μM W130Q at 42 °C; *Purple diamonds*, 50 μM wild-type isolated N-terminal domain of HγD at 44 °C. All experiments were carried out in 10 mM PIPES pH 6.7, 150 mM NaCl, 1 mM EDTA, with 0.5 mM GSSG as the oxidant. Raw data and fitting for panels C,D,E are available as *Figure 1—source data 2*.

The online version of this article includes the following source data and figure supplement(s) for figure 1:

**Source data 1.** Raw numerical values for turbidity endpoint assays in *Figure 1B*.

**Source data 2.** Raw numerical data and fittings for solution turbidity data in *Figure 1C,D,E*.

**Figure supplement 1.** 100 mM inositol does not significantly affect bulk solution properties.

**Figure supplement 2.** Comparison of *myo*- and *scyllo*-inositol.

*Figure 1 continued on next page*

*Figure 1 continued*

**Figure supplement 3.** Raman spectroscopy shows no signature of increased inositol self-association between 100 and 500 mM.

**Figure supplement 4.** Aggregation of W130Q and isolated WT N-terminal domain is completely redox-dependent.

maximum rate, lag time, and endpoint turbidity (*Serebryany et al., 2019*). It has been estimated that delaying the age of onset of cataract by ~14% would halve its global disease burden (*WHO, 2014*); conversely, the near-complete loss of *myo*-inositol in cataractous lenses (*Tsentalovich et al., 2015*; *Yanshole et al., 2019*) could be a major factor increasing that burden. Less-abundant lens metabolites, including *scyllo*-inositol and glucose, likely contribute to aggregation suppression. Glycerol, previously shown to be also abundant in the lens (*Tomana et al., 1984*), was much less effective. *Scyllo*-inositol offered no advantage over *myo*-inositol (*Figure 1—figure supplement 2*) and was not pursued further owing to its lower solubility and higher cost. In crystals, *myo*-inositol self-associates via networks of hydrogen bonds (*Rabinovich and Kraut, 1964*). Although the maximum *myo*-inositol concentrations used in this study were not far below the empirical solubility limit of ~500 mM, Raman spectroscopy (*Figure 1—figure supplement 3*) did not reveal any signature of direct self-association in the 100–500 mM concentration range. However, as with any alcohol, transient water-mediated clusters (*Dolenko et al., 2015*) cannot be ruled out for inositol.

We next checked whether aggregation suppression was broadly effective or specific to the W42Q HγD variant. Dose-dependent suppression of oxidative aggregation was observed in a variety of aggregation-prone HγD constructs. We have previously found that, while the W42Q variant aggregates on its own at sufficient concentrations, the wild-type protein promotes W42Q aggregation without itself aggregating (*Serebryany and King, 2015*; *Serebryany et al., 2018*). *Figure 1E* shows that aggregation of low [W42Q] in the presence of high [WT] was also suppressed by *myo*-inositol. The W130Q variant, which mimics oxidation of the cognate tryptophan in the C-terminal domain, was likewise rescued. The N-terminal domain of HγD derives part of its stability from the domain interface, which may be disrupted over time, for example, by deamidation or truncation (*Flaugh et al., 2006*; *Mills et al., 2007*); the isolated domain forms aggregates at slightly elevated temperatures with oxidation (*Serebryany et al., 2019*). Inositol suppressed aggregation of the wild-type N-terminal domain of HγD ('N-only') like that of the other variants. Aggregation of N-only and W130Q, like that of W42Q, was entirely redox-dependent (*Figure 1—figure supplement 4*), so the aggregation precursor conformations adopted by all these variants are likely kinetically trapped by non-native disulfide bonds. Since the two Cys residues forming the most aggregation-favoring disulfide (Cys32 and Cys41) are highly conserved among γ-crystallins (*Serebryany et al., 2021*), we expect that this aggregation pathway is typical for the γ-crystallin family and predict that *myo*-inositol and its isomers and derivatives suppress aggregation of other γ-crystallins, whether they are damaged by mutations or by covalent side-chain modifications or backbone truncations accumulated during the course of aging. The rest of our study concerns the mechanism of aggregation suppression by *myo*-inositol.

## Inositol neither binds the native state nor destabilizes the unfolded state

The effects of chemical chaperones (protective osmolytes) on proteins are typically conceptualized as increasing the protein's thermodynamic stability, either by stabilizing the native state via a binding interaction or by destabilizing the unfolded state via preferential protein backbone exclusion (*Dandage et al., 2015*; *Okiyoneda et al., 2013*; *Street et al., 2006*). We obtained $^1$H-$^{15}$N HSQC NMR spectra of WT and W42Q HγD at 37 °C with and without 100 mM inositol (*Figure 2A and B*). This inositol concentration is in the estimated physiological range and clearly suppresses aggregation (see above). From the underlying data of *Figure 1A and B*, we calculated maximum aggregation rate suppression of 35% ± 3% (mean ± S.E.M. of four experiments). Yet, the NMR spectra for each protein with and without inositol were superimposable; the minor systematic shift with inositol does not indicate specific binding, only a change in the solvent environment due to addition of osmolyte (*Iwaya et al., 2020*). Chemical shift distributions are shown in *Figure 2—figure supplement 1*. In sum, we found no defined binding interaction between inositol and the native structure of HγD WT or W42Q.

If *myo*-inositol's effect were achieved by destabilizing the unfolded state, then it would increase native thermodynamic stability even without binding the native state. To improve assignment of

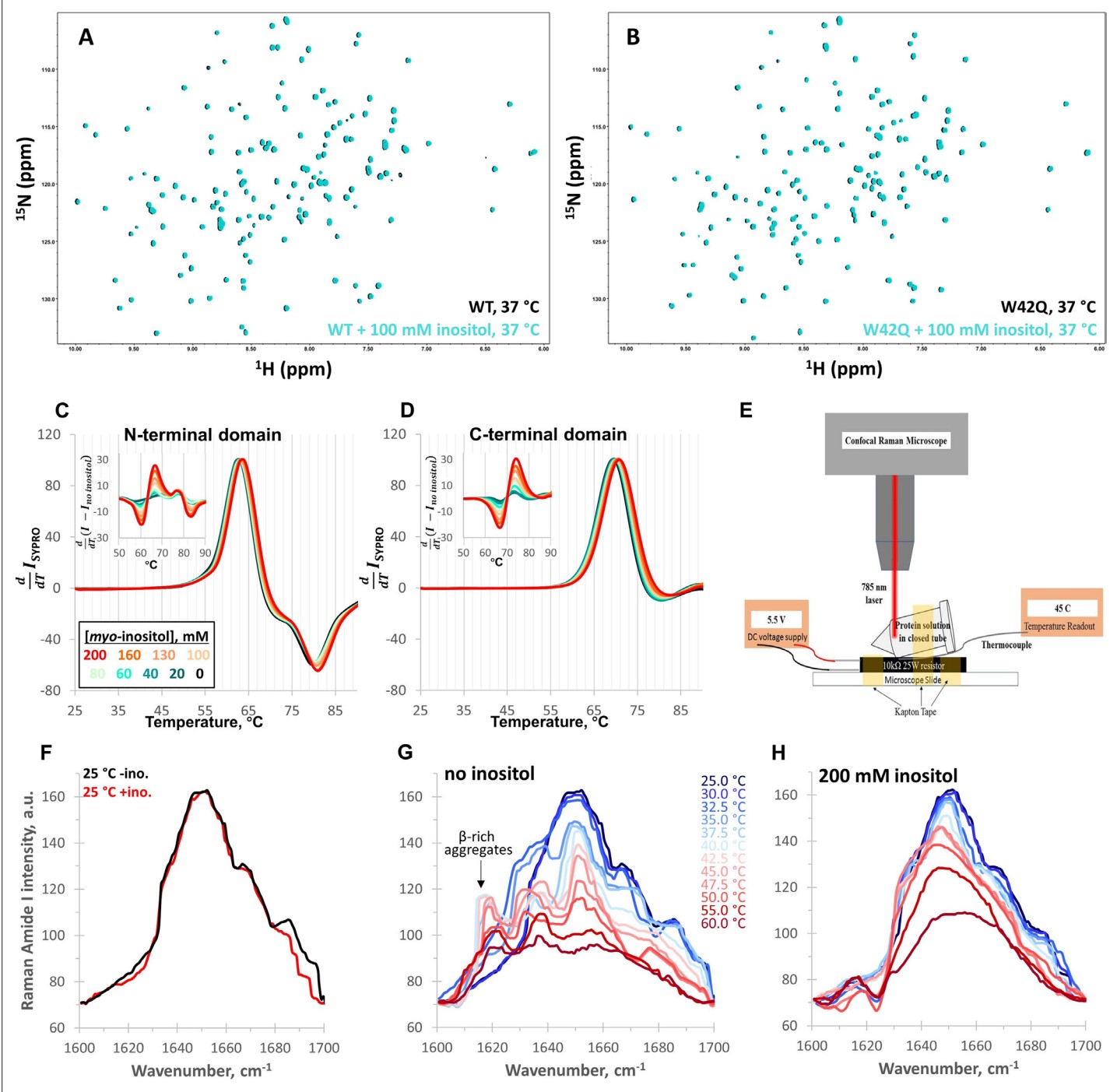

**Figure 2.** *Myo*-inositol has no effect on the native structure and only minor effects on stability. (**A, B**) $^1$H-$^{15}$N HSQC NMR spectra of full-length WT and W42Q HγD at 37 °C showed no evidence of interaction between *myo*-inositol and the protein's native structure. (**C, D**) Differential scanning fluorometry with SYPRO Orange as the hydrophobicity probe revealed no change within the temperature range of aggregation but very small dose-dependent shifts toward higher melting temperatures. Normalized first derivative curves of SYPRO Orange fluorescence intensity are shown for both the N- and C-terminal domains, where peak positions correspond to $T_m$; insets show the difference curves. (**E**) Schematic of the design of the thermal scanning Raman spectroscopy apparatus. (**F**) Raman Amide I band spectra of the isolated N-terminal domain were likewise overlapping at low temperatures. (**G, H**) At 32.5 °C and 35.0 °C without inositol, a β-sheet peak ~1630 cm$^{-1}$ became prominent, while a ~1620 cm$^{-1}$ peak, typical of β-sheet-rich aggregates, was observed above 35.0 °C. Both features were strongly suppressed by 200 mM inositol.

The online version of this article includes the following figure supplement(s) for figure 2:

**Figure supplement 1.** Histograms of chemical shift perturbations in WT and W42Q HγD upon addition of 100 mM *myo*-inositol.

*Figure 2 continued on next page*

*Figure 2 continued*

**Figure supplement 2.** Secondary structure content of the WT and N terminal domain obtained upon deconvolution of the Amide I Raman spectra was comparable to the reported crystal structure (1HK0).

unfolding transitions, we measured thermal stability for each HγD domain separately as a function of [inositol] by differential scanning fluorometry (DSF) with SYPRO Orange as the hydrophobicity probe (*Figure 2C and D*). Only very minor stabilization was observed (up to ~1 °C at the highest [inositol]) – not sufficient to account for the aggregation suppression.

If inositol were to bind and stabilize a partially unfolded state, it would decrease the native state's stability. This was not observed. Nevertheless, since the DSF traces were silent at the N-terminal domain's aggregation-permissive temperature (44 °C in *Figure 1E*), we sought a more sensitive measurement of any structural changes in that range. We therefore constructed a thermal scanning Raman spectroscopy apparatus (schematized in *Figure 2E*). Protein solutions in sealed vessels (to avoid evaporation) were placed under a Raman microscope. We scanned the entire amide I region (1600–1700 $cm^{-1}$), which reports mostly (70–85%) the C=O stretch, with a smaller contribution from the C-N stretch, and is therefore an excellent probe of protein secondary structure (*Williams, 1983*; *Williams and Dunker, 1981*). Overall signal strength is a rough measure of foldedness.

Even at 200 mM [*myo*-inositol], the Amide I spectra of N-only at 25 °C with and without inositol overlapped, except for a very small feature at ~1690 $cm^{-1}$ that could not be confidently assigned (*Figure 2F*), and the calculated secondary structure content under both conditions matched the PDB reference structure (*Figure 2—figure supplement 1*). Temperature was then gradually ramped from 25°C to 60 °C to investigate the early unfolding transitions (*Figure 2G and H*). In this experiment, inositol clearly suppressed conformational change in the sample (*Ganim et al., 2008*), notably the emergence of a ~1620 $cm^{-1}$ peak (*Figure 2G*), which was attributable to β-sheet-rich aggregates (*Holzbaur et al., 1996*). Sample with inositol, unlike the control, lost signal intensity in smooth progression during the temperature ramp (*Figure 2H*), indicating loss of structure without misfolding. A distinct feature in the β-sheet range (*Byler and Susi, 1986*), at ~1630 $cm^{-1}$, became prominent in the absence of inositol at T=32.5 and T=35 °C, before the ~1620 $cm^{-1}$ peak emerged, raising the possibility that these were early-stage aggregates or precursors that inositol suppressed.

## Inositol produces only subtle changes in aggregate morphology and particle size distribution

We used negative-stain transmission electron microscopy to investigate the effect of *myo*-inositol on the aggregation process. Notably, even the control sample contained a small number of large condensed aggregate particles (*Figure 3A*); these particles form during storage and incubation without reducing agent and contribute very little to overall turbidity. Samples from the turbid solutions (*Figure 3B and C*) showed much greater texturing on the survey micrographs and were replete with smaller aggregates. *Figure 3D–G* show representative images of the observed types of aggregates and suggest a sequence of structural transitions on the way to the insoluble aggregated state (*Figure 3H*). *Myo*-inositol produced no qualitative change in the morphology of the aggregates.

We next carried out morphometry of the aggregates, using a total of 89 separate images from duplicate aggregated samples with 0, 100, or 250 mM *myo*-inositol plus 14 from the control (non-GSSG-treated) sample. A double-blind was set up to minimize human bias: the microscopist did not know which sample came from which treatment condition, and the image analyst did not know which images came from which sample. Aggregates were visually classified into globular/collapsed or extended/fibril-like based on whether a curve could be clearly traced from one end to the other; aggregates composed of multiple extended chains or featuring intra-chain interactions that obscured one or both ends were considered globular. For this reason, the extended aggregates were typically short in length.

We used two alternative quantification approaches: the size distributions of particles ranked from largest to smallest (*Figure 4A and D*) and statistics of the size and number of aggregates of either type by image (*Figure 4B and C*; *E and F*). Inositol-induced changes in aggregate size distributions were minor but mechanistically informative. At the intermediate, near-physiological concentration (100 mM), the extended aggregates became shorter and the globular ones fewer in number. By contrast, globular aggregates in the intermediate range (rank ~25–55 in *Figure 4D*) grew slightly

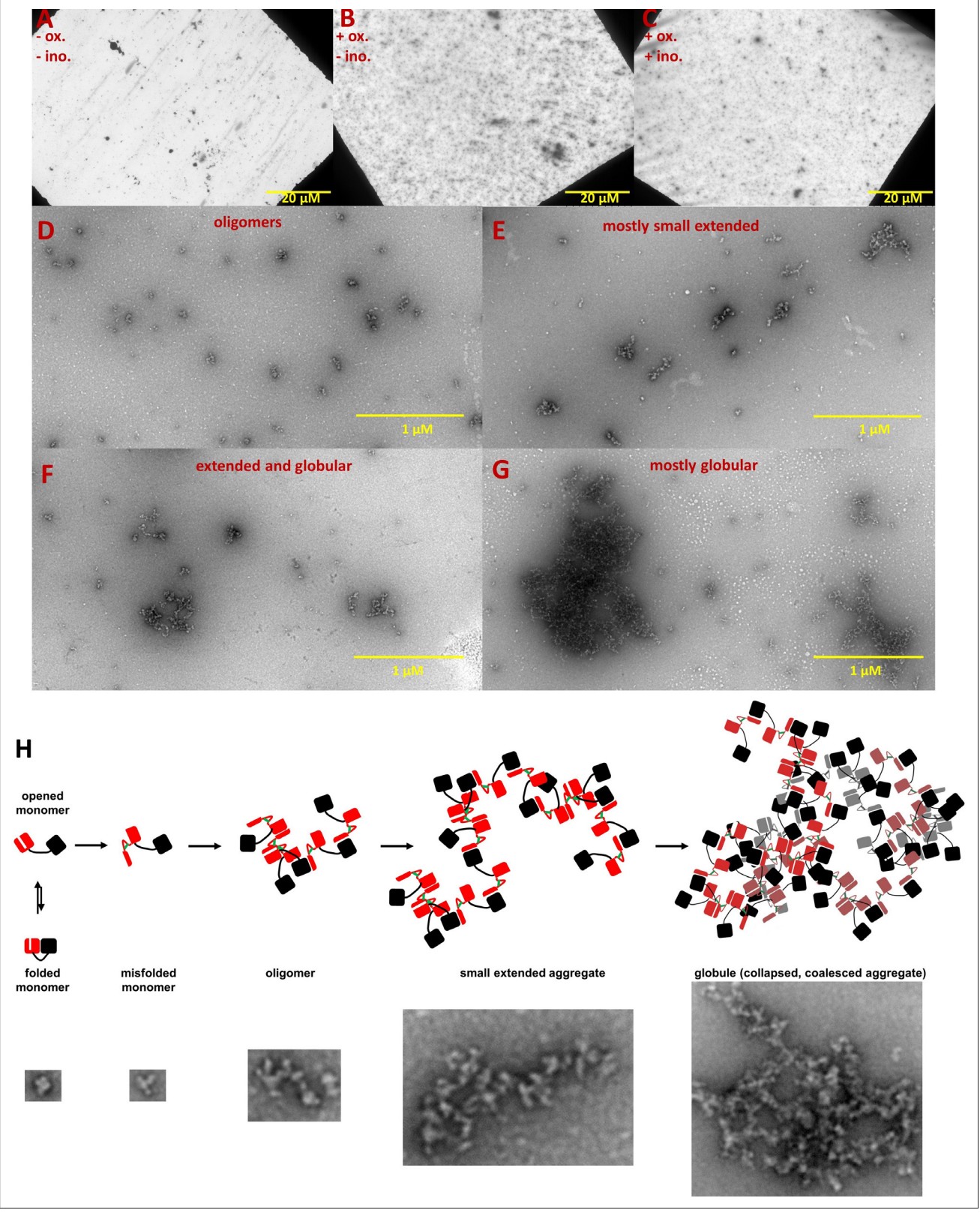

**Figure 3.** Negative-stain TEM images of HγD W42Q aggregates reveal short extended and large globular structures. Aggregation of 40 μM protein in pH 6.7 PIPES buffer with 150 mM NaCl and 1 mM EDTA was triggered by addition of oxidant (0.5 mM GSSG) and incubation at 37 °C for 4 hr. End-point samples were deposited on carbon-coated copper grids and stained with uranyl acetate. The top row shows survey images of whole grids for (**A**) control sample incubated in the absence of GSSG; (**B**) turbid sample in the presence of GSSG; and (**C**) turbid sample in the presence of GSSG and 100 mM

*Figure 3 continued on next page*

*Figure 3 continued*

*myo*-inositol. Panels D-G show the variety of aggregate morphologies observed in representative magnified images from these grids, with (**D**) showing mostly small oligomers (not counted as aggregates in *Figure 4*); (**E**) showing mostly small extended aggregates; (**F**) showing extended aggregates in the process of collapsing and coalescing; and (**G**) showing highly coalesced globular aggregates; (**H**) Graphical model of HyD aggregation and its suppression by *myo*-inositol: aggregation begins with oxidative misfolding of a mutant or damaged protein, as we have previously found (**Serebryany et al., 2016b**), followed by assembly of short extended chains via domain swap-like interactions (**Serebryany et al., 2016b**; **Serebryany et al., 2018**), and finally coalescence to globular particles. Examples of TEM evidence of each type of structure in this study are shown in the bottom row (not all to scale).

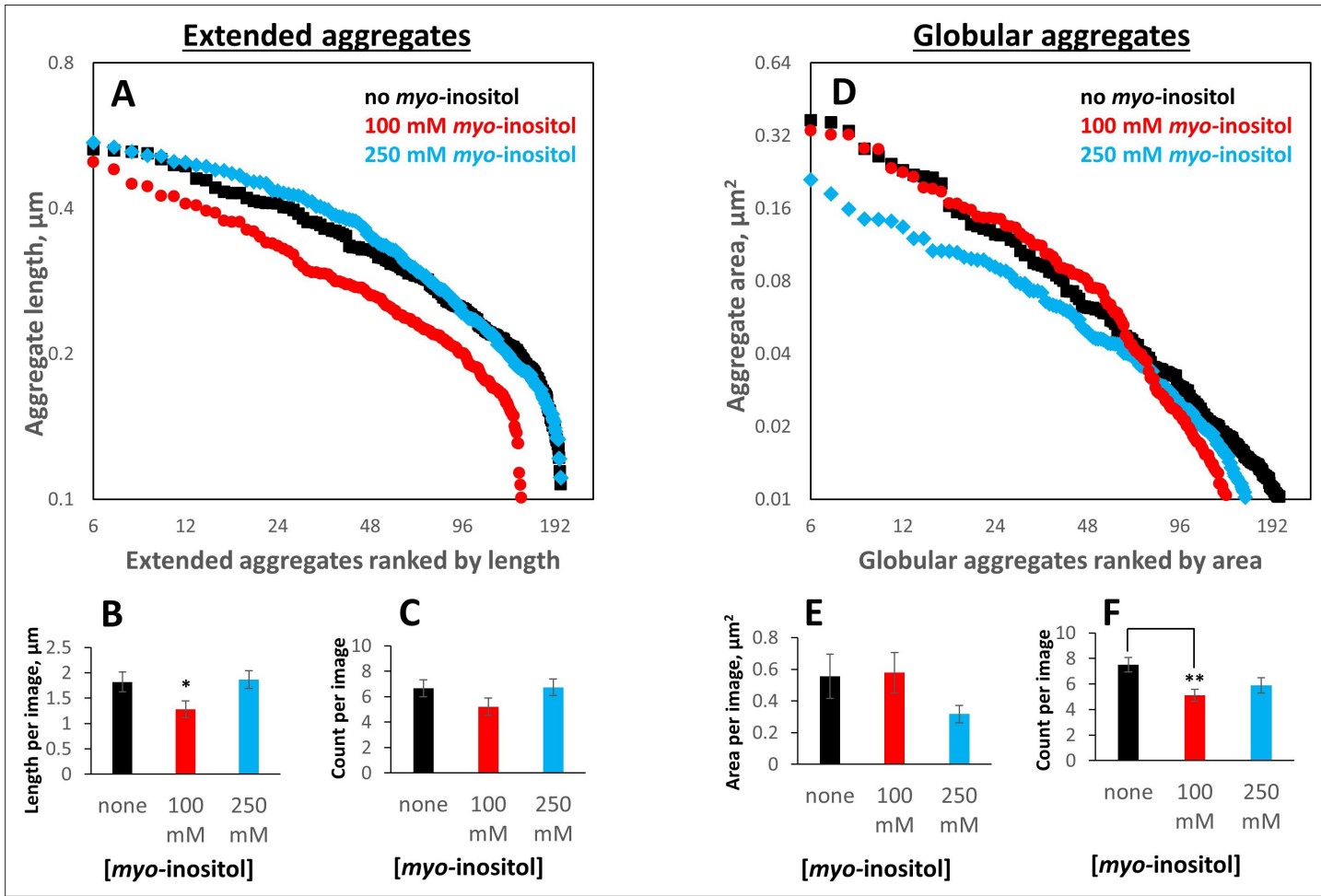

**Figure 4.** *Myo*-inositol produces only modest shifts in the aggregate particle size distribution. Aggregates generated as in *Figure 1* in the presence of 0, intermediate (100 mM) or high (250 mM) *myo*-inositol were imaged and analyzed by negative-stain transmission electron microscopy (*Figure 3*) in a double-blinded procedure, with a total of 103 separate images used for analysis (30, 29, and 30 for 0, 100, and 250 mM inositol, respectively, plus 14 control images). Aggregates were classified by inspection into extended (irregular fibrils) vs. globular (collapsed/coalesced) based on whether a curve could be unambiguously traced from end to end. (**A, B, C**) 100 mM *myo*-inositol decreased both the number and the length of extended aggregates, but 250 mM *myo*-inositol did not. Error bars represent S.E.M. of particle counts and their combined lengths per image. Two-sample T-test p=0.04 and 0.02 between 0 and 100 mM and 100 and 250 mM, respectively, in (B) and 0.13 and 0.11, respectively, in (C). (**D, E, F**) 100 mM *myo*-inositol depleted the smallest globular aggregates, slightly increased the size of intermediate ones, and left the largest ones unchanged, but 250 mM *myo*-inositol reduced the size of the largest aggregates. Two-sample T-test p=0.12 and 0.07 between 0 and 250 mM and 100 and 250 mM, respectively, in (E), and p=0.003 between 0 and 100 mM in (F).

The online version of this article includes the following source data for figure 4:

**Source data 1.** Raw numerical data for aggregate particle size distributions.

larger than without the inhibitor. By contrast, at the high *myo*-inositol concentration (250 mM), the extended aggregates lengthened slightly compared to the no-inositol control (*Figure 4A*), yet the larger, globular aggregates shrank across the board (*Figure 4D*). The shapes of the size distributions for the globular aggregates under the three treatment conditions (*Figure 4D*) were compared by two-sided Kolmogorov-Smirnov tests. These tests indicated statistically likely differences between 0 and 100 mM *myo*-inositol (p=0.0032) and between 100 and 250 mM *myo*-inositol (p=0.0048). The shapes of the aggregate size distributions at 0 and 250 mM *myo*-inositol were not statistically distinct (p=0.22), though the distribution of aggregate sizes at 250 mM was clearly shifted downward.

While *myo*-inositol did not appreciably alter overall aggregate morphology, the more subtle shifts in aggregate size distributions presented a seeming paradox. At 100 mM, the compound reduced the number and size of smaller aggregates but not of larger ones, while at 250 mM it decreased the size of the larger aggregates more than the smaller ones (indeed, the smaller extended aggregates appeared to lengthen). We can rationalize this behavior by observing that large globular aggregates frequently contained internal cavities, which suggests they formed via condensation and coalescence of smaller ones (*Figure 3H*). *Myo*-inositol may slow the earliest steps of aggregate assembly, up to the formation of short extended chains, which assemble predominantly one protein-protein interaction at a time. Collapse and coalescence of the small aggregates involve many protein-protein interactions simultaneously, and the high avidity of interactions, along with possible formation of rare intermolecular disulfides, could overcome the aggregation suppression by low or intermediate [*myo*-inositol]. High [*myo*-inositol], however could slow down the early aggregation steps to such a degree as to effectively shift the entire aggregate size distribution to an earlier point on the aggregation master curve, consistent with the increased lag time (*Figure 1A and D*). Essentially, the largest aggregates may not have had time to form by the endpoint of the assay, leaving a relatively larger proportion of the aggregates small.

Although 100 mM inositol (1.8% w/v) did not change bulk solvent properties (*Figure 1—figure supplement 1*), we found that the solubility limit of *myo*-inositol was ~500 mM in water or aqueous buffer, so concentrations in the high-millimolar or molar range could not be assayed. The aggregate particle size distributions are tabulated in *Figure 4—source data 1*, and all raw TEM images are available at: https://doi.org/10.7910/DVN/BVRS9M.

## Inositol inhibits the rate-limiting bimolecular step and slows formation of disulfide-trapped aggregation precursors

The redox potential of lens fiber cell cytoplasm becomes progressively more oxidizing with age (*Serebryany et al., 2021*). While all experiments in *Figures 1–4* were carried out in fully oxidizing buffers

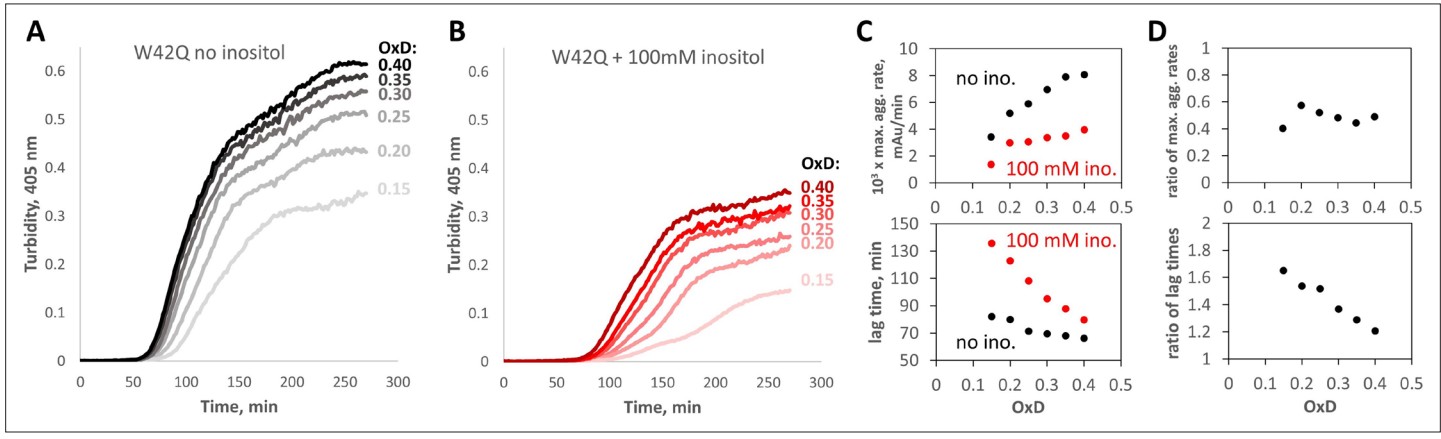

**Figure 5.** *Myo*-inositol suppresses aggregation more strongly in redox buffers. (**A, B**) Turbidity traces of 50 µM W42Q in buffers with set redox potentials using the glutathione redox couple, where OxD = 2[GSSG]/([GSH]+2[GSSG]), and ([GSH]+2[GSSG])=2 mM in all cases. The OxD range shown is consistent with what is observed during cataract development (*Hains and Truscott, 2007*). Notably, in the least-oxidizing buffers with *myo*-inositol, the turbidity curves became biphasic. (**C**) Quantification of the maximum aggregation rates and lag times for the curves in (**A**) and (**B**) showed stronger aggregation suppression by 100 mM *myo*-inositol than in *Figure 1A and B*. (**D**) Ratios of maximum aggregation rates (+inositol/-inositol) (*top*) and aggregation lag times (*bottom*), for the data shown in panel (**C**). Averaged across the six OxD buffers, mean ± S.E.M. for inositol's inhibitory effect on the aggregation rate was 51±2% vs 35 ± 3% observed in the fully oxidizing buffer.

(0.5 mM GSSG), we have previously shown that W42Q and W42R aggregation proceeds in the physiologically relevant range of redox-coupled buffers (*Serebryany et al., 2016b*). *Myo*-inositol is redox-inert, but we nonetheless investigated whether its aggregation suppressing effect was affected by the degree of oxidation of the buffer. We defined degree of oxidation (OxD) as it was used by *Romero-Aristizabal et al., 2014*, as follows:

$$OxD = \frac{2[GSSG]}{([GSH]+2[GSSG])}$$

Note that the true redox potential of the buffer depends not only on OxD but also on the total concentration of glutathione, since the formation of glutathione disulfide is bimolecular (*Romero-Aristizabal et al., 2014*). We used physiologically plausible total glutathione of 2 mM in most of the following experiments, except in Figure 6 as noted.

As shown in *Figure 5*, aggregation at OxD 0.15–0.40 was OxD-dependent but always more strongly suppressed by inositol than in the fully oxidizing buffers (*Figure 1A and B*). (Overall, the aggregation propensity in these experiments was higher than in our previous study *Serebryany et al., 2016b*; this difference is attributable to higher buffer ionic strength.) Suppression of maximum aggregation rate did not vary significantly among this series of redox buffers; we observed maximum aggregation rate suppression of 51% ± 2% (mean ± S.E.M. for the six redox buffers), significantly greater than the 35% ± 3% obtained from the fully oxidizing buffers (*Figure 1*). By contrast, the apparent aggregation lag times became significantly longer at low OxD values in the presence of inositol (*Figure 5C and D*). This was consistent with the above evidence that inositol suppressed an early stage of aggregate formation. In fact, low OxD combined with 100 mM inositol slowed early (but not late) aggregate assembly so much that turbidity traces became visibly biphasic (*Figure 5B*). Since both the maximum rate and the lag time were calculated from the second phase (see *Figure 6—figure supplement 2* for illustration), the apparent lag times became significantly longer.

We have previously shown that, while aggregation requires disulfide-trapped misfolding of the N-terminal domain, the two Cys residues in HγD's C-terminal domain, which are not directly involved in the aggregation process, constitute an oxidoreductase site (*Serebryany et al., 2018*). To investigate the effect of inositol on the redox state distribution most relevant to the aggregation process, we removed the oxidoreductase site from the W42Q variant, generating the triple-variant W42Q/C108S/C110S. End-point samples of W42Q/C108S/C110S aggregates at various OxD's were then reacted with PEG5000-conjugated maleimide. The gel shifts due to PEGylation are an excellent reporter of the redox state distribution because, unlike the commonly used Ellman assay, they report not just the average number of free thiols per molecule but the full distributions of the number of free thiol groups per protein molecule (*Serebryany et al., 2018*). Since the maleimide group needed to be in excess of thiols to ensure complete labeling, total glutathione was lowered to 0.5 mM for these experiments. Under those conditions, 200 mM inositol suppressed aggregation of the triple-variant almost completely (*Figure 6A*).

In the more mildly oxidizing buffers (OxD' 0.15 or 0.20), a clear shift in the redox state distribution was observed in the presence of inositol (*Figure 6B*). The quadruply-PEGylated band, which corresponds to fully reduced protein, became visibly more intense, while intensity of the bands corresponding to one or two internal disulfides per protein molecule weakened relative to the no-inositol control. (Full gels are shown in *Figure 6—figure supplement 1*). The samples for PEGylation were taken from the bulk solution without mixing, so any aggregates large enough to settle to the bottom of the reaction well were not collected. We have previously shown that W42Q aggregation requires a non-native disulfide in each protein molecule, so the aggregates do not contain any fully reduced protein (*Serebryany et al., 2018*); the lower intensities of quadruply-PEGylated bands without inositol in *Figure 6B* are attributable to lower amounts of the protein remaining soluble at the endpoint of the assay. Thus, surprisingly, *myo*-inositol was able to shift the molecular population in bulk solution toward the reduced state, despite not being itself redox-active. We therefore propose a mode of action for inositol schematized in *Figure 6C*: inhibition of the rate-limiting bimolecular step effectively drives the early-stage aggregation process in reverse when a redox buffer is present. We note that HγD itself has some capacity to serve as a redox buffer (*Serebryany et al., 2018*).

We have previously shown that the concentration-dependence of W42Q aggregation rate is quadratic, indicating a bimolecular rate-limiting step (*Serebryany et al., 2019*; *Serebryany and King,*

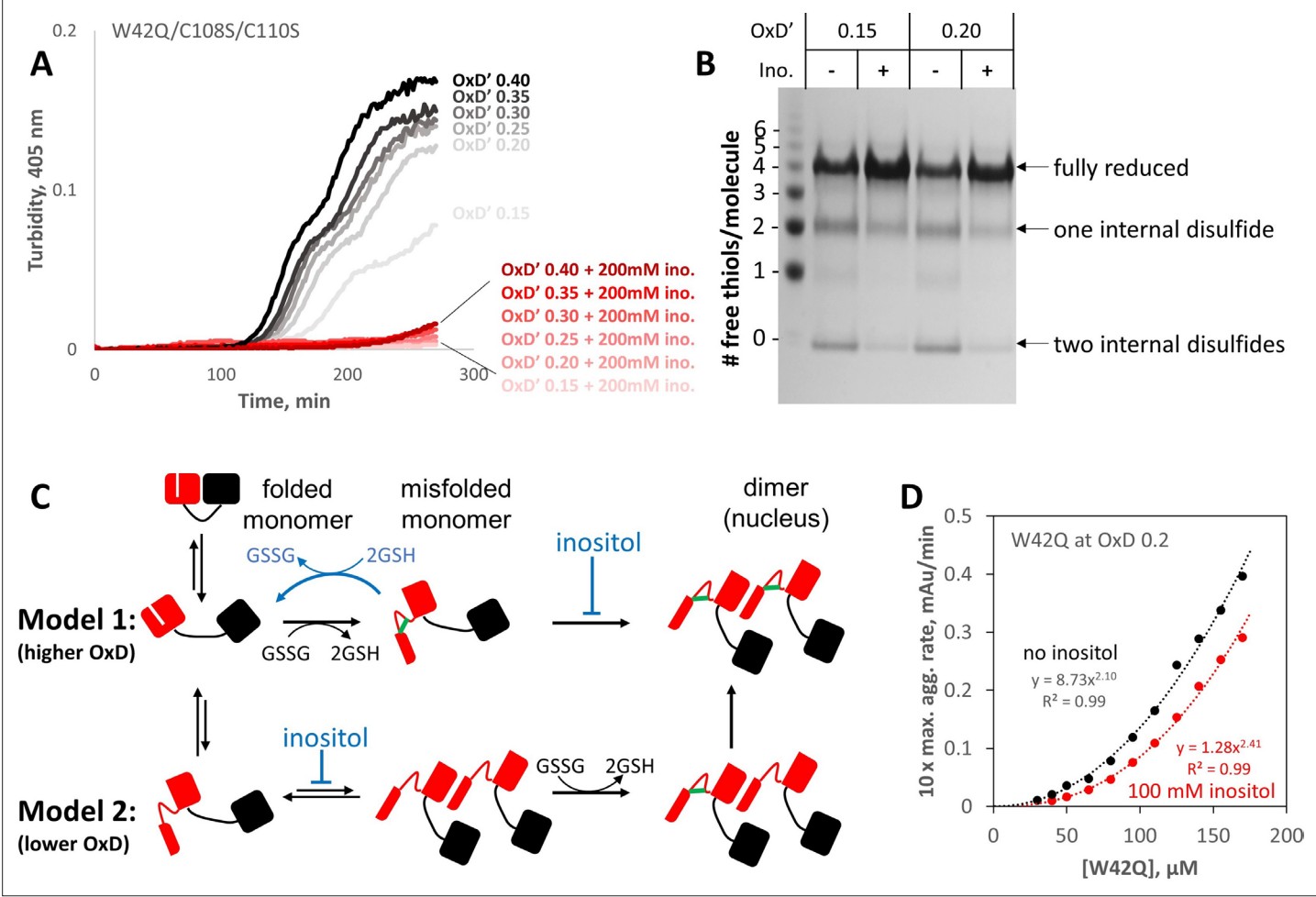

**Figure 6.** *Myo*-inositol targets the rate-limiting bimolecular interaction and indirectly favors reduction of disulfide-locked misfolded aggregation precursors. (**A**) Aggregation of the W42Q/C108S/C110S triple-mutant (lacking the redox-active site in the C-terminal domain) was almost completely suppressed by 200 mM *myo*-inositol. Here ([GSH]+2[GSSG]) was only 0.5 mM to facilitate subsequent reaction of the protein with PEG-maleimide; as such, the redox buffers are designated OxD' to distinguish them from those in *Figure 5*. (**B**) PEGylation gel-shift assays of the end-points of aggregation reactions in panel A reveal the redox state distributions of the protein, where four free thiols per molecule indicate it is fully reduced; two free thiols indicate one internal disulfide; and 0 free thiols, two internal disulfides. Markers were generated by limited PEGylation of the WT protein, which contains six thiols. The differential effect of inositol was maximal at OxD' 0.15 and 0.2; results for higher and lower OxD' values are shown in the figure supplement. (**C**) Two alternative graphical models explaining how *myo*-inositol, despite being redox-inert, can alter the redox state distribution indirectly in redox-buffered solutions. Models 1 and 2 are not mutually exclusive and differ from each other by the order of misfolding and oxidation: high and low OxD is expected to favor Model 1 and Model 2, respectively. When disulfide bonding is reversible, inositol may favor reduction and refolding by suppressing the transition from misfolded monomer to transient dimer. (**D**) Concentration dependence of W42Q aggregation at OxD 0.2 with and without 100 mM *myo*-inositol. The rightward shift in the curve with inositol (smaller pre-exponential factor) suggests that inositol makes it less likely that a bimolecular interaction will be productive for aggregation, while the larger exponent in the power law further indicates that in the presence of inositol assembly beyond the dimer makes a greater relative contribution to aggregation. Raw data and fitting are available as *Figure 6—source data 1*.

The online version of this article includes the following source data and figure supplement(s) for figure 6:

**Source data 1.** Raw numerical data and fittings for solution turbidity experiments in *Figure 6D*.

**Figure supplement 1.** PEGylation gel shift assays reveal shifts in the redox distribution of the W42Q/C108S/C110S protein induced by 200 mM *myo*-inositol.

**Figure supplement 1—source data 1.** Full unedited images of gels in *Figure 6—figure supplement 1*.

**Figure supplement 2.** Concentration-dependence of W42Q aggregation curves with and without *myo*-inositol.

*2015*). The proposed model in *Figure 6C* predicts that, by inhibiting this rate-limiting step, *myo*-inositol could alter the concentration dependence of W42Q aggregation in two ways. First, if it makes a dimer less likely to contribute to aggregation, then it should decrease the pre-exponential factor. Second, if it requires an assembly larger than a dimer to initiate or propagate an aggregate (e.g. a trimer is now required to attain the requisite stability), then it should increase the exponent of the concentration dependence power law. We observed both these effects in the [W42Q]-dependence experiment in *Figure 6D*.

At the lowest [W42Q], both with and without inositol, the solution turbidity traces became biphasic (*Figure 6—figure supplement 2*), as they did at the lowest OxD values with inositol (*Figure 5B*), while the triple-mutant showed biphasic aggregation throughout (*Figure 6A*). While a thorough investigation of that behavior is beyond the scope of the current study, the most likely hypothesis is that early-stage aggregation is dominated by formation and growth of the aggregates, while later stage (second phase) aggregation is dominated by coalescence of the smaller aggregate particles to form larger ones, as schematized in *Figure 3H*. The second phase appeared to be smaller for W42Q/C108S/C110S than for W42Q (compare *Figure 6A* OxD' 0.2 vs. *Figure 5B* OxD 0.2 and *Figure 6—figure supplement 2*),

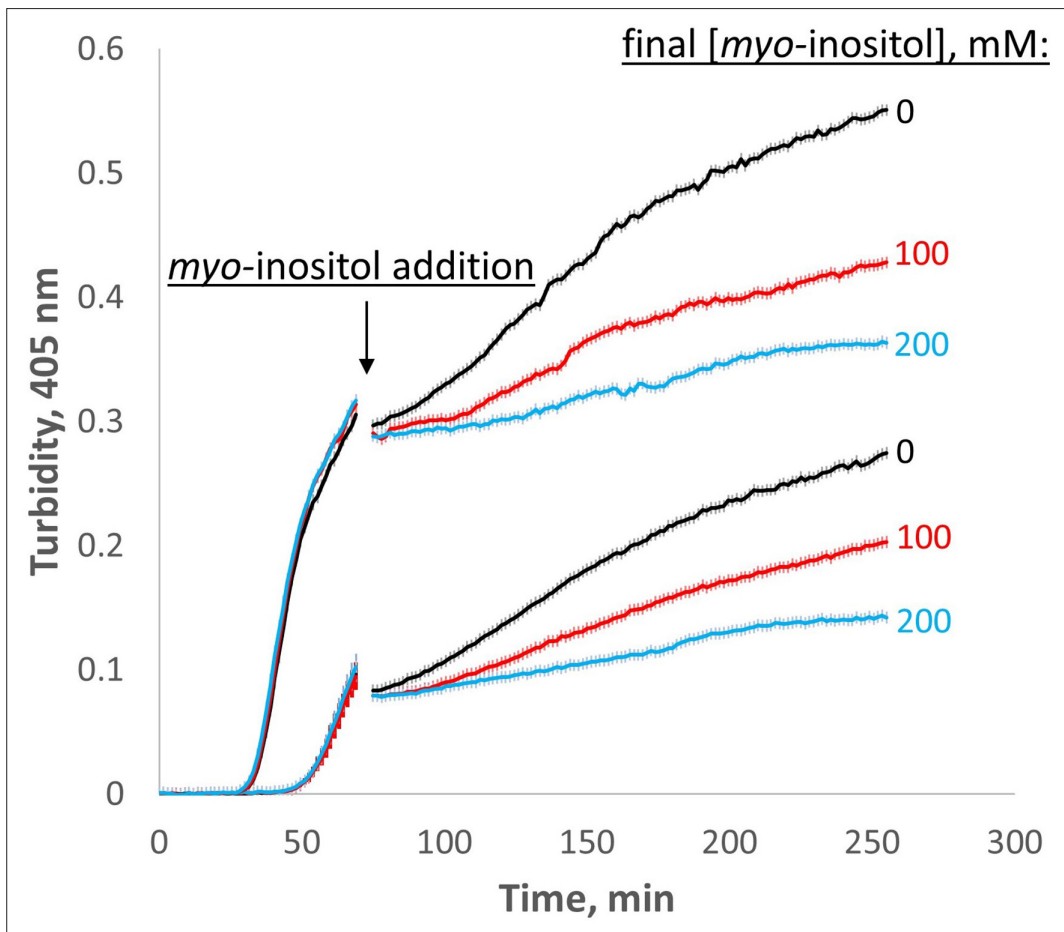

**Figure 7.** *Myo*-inositol inhibits aggregation even at later stages. When added late in the aggregation process, *myo*-inositol slowed down the rate of subsequent aggregation in a dose-dependent manner. Triplicate W42Q samples were allowed to aggregate initially in 60 μl volume, then 40 μl of *myo*-inositol solutions or blank buffer was added and the incubation resumed. Light scattering did not change greatly on dilution due to the vertical optics of the plate reader. The discontinuity in the curves shows the dead time of *myo*-inositol addition and mixing. The curves shown are averages of four technical replicates at each of two different [W42Q] (*upper curve*, 75 μM; *lower curve*, 50 μM) in OxD 0.2 buffer, with error bars showing S.E.M. for each turbidity reading (every 90 s).

The online version of this article includes the following figure supplement(s) for figure 7:

**Figure supplement 1.** *Myo*-inositol extends the first (slow) phase of aggregation when added early in the aggregation process.

perhaps because the latter may form disulfides between the solvent-exposed C-terminal Cys residues of subunits from different aggregate particles in addition to non-covalent coalescence.

Our finding that *myo*-inositol primarily targets the early, bimolecular step on the aggregation pathway raises the question of whether it can have an effect when added later in the process, after the initial aggregates have formed. This question has important implications for whether inositol could potentially serve as a treatment for cataract or only as a prophylactic. The model in *Figure 6C* predicts that inositol should inhibit the growth of aggregates even when it is added at a later stage because dimerization is important for generating or stabilizing the misfolded conformer that is required for continued aggregation. (In the extreme case, aggregate growth at later stages may even proceed via dimer rather than monomer addition.) We therefore carried out experiments in which W42Q was allowed to aggregate initially at OxD 0.2 as in *Figure 5A*, followed by addition of inositol after the rapid phase of the aggregation (*Figure 7*). The initial aggregation was somewhat more rapid than the comparable traces in *Figure 5A* and *Figure 6—figure supplement 2*, respectively, perhaps due to the greater surface area:volume ratio of the initial aggregation reactions; interfaces, including the air-water interface, are known to promote protein misfolding. Nonetheless, when *myo*-inositol was added to the already-turbid solutions, the rate of further turbidity development became substantially and reproducibly slower than the blank-buffer controls. Thus, *myo*-inositol can still serve as an aggregation suppressor even at a later stage.

We also carried out a set of experiments with an alternative protocol, in which [W42Q] was kept sufficiently low to generate biphasic aggregation and inositol (or blank buffer) was added during the first phase of the aggregation process, along with enough fresh W42Q to keep [W42Q] constant (*Figure 7—figure supplement 1*). In these experiments, mixing produced a significant drop in turbidity, consistent with our prior findings that very-early stage aggregates are fragile (*Serebryany and King, 2015*). In all cases, *myo*-inositol prolonged the first (slow) aggregation phase, consistent with our model that its primary effect is on the dimer and (thereby) on precursor availability. Curves in this experiment became reproducibly multiphasic, pointing to a complex interplay among growth of pre-existing aggregates, formation of new ones from pre-existing precursors, and eventual misfolding of newly added W42Q.

## Discussion

Natural chemical chaperones, including *myo*-inositol, are typically expected to stabilize the native state of a protein, either directly by binding it or indirectly by destabilizing the unfolded state (*Beg et al., 2017*; *Clark et al., 2012*; *Dandage et al., 2015*; *Dong et al., 2019*; *Gault et al., 2018*; *Gupta et al., 2016*; *Morgan et al., 2019*; *Okiyoneda et al., 2013*; *Street et al., 2006*). In some cases, osmolytes have been found to alter aggregate morphology (*Bashir et al., 2020*; *Marasini et al., 2017*). Inhibitors of aggregation kinetics whose primary target is transient oligomeric states or nuclei have been more challenging to find, let alone characterize, but they are of especially high interest as possible treatments for proteinopathies (*Giorgetti et al., 2018*; *Habchi et al., 2016*; *Ignatova and Gierasch, 2006*; *Petrosyan et al., 2021*).

Given the strong evolutionary pressure to preserve clear vision, metabolites serving the chemical chaperone function should be expected in eye lenses across much of the animal kingdom. Thus, while human lenses are rich in *myo*-inositol, rat lenses have less *myo*-inositol but much more taurine, another potential chemical chaperone (*Kinoshita et al., 1969*; *Yanshole et al., 2014*). Fish lenses are relatively low in both *myo*-inositol and taurine but highly enriched in N-acetyl histidine instead (*Rhodes et al., 2010*; *Tsentalovich et al., 2019*). Pharmacological treatment of cataract disease is particularly challenging because the vast majority of even small compounds fail to permeate into this dense and diffusion-limited tissue (*Heikkinen et al., 2019*). Natural aggregation-suppressing metabolites could thus be a useful strategy. *Myo*-inositol, which readily enters the lens thanks to dedicated transporters (*Cotlier, 1970*; *Diecke et al., 1995*), had previously been thought of as an osmotic regulator in the lens (*Zhou et al., 1994*). We have now demonstrated that this compound, in the physiological range of its healthy lenticular concentrations, can suppress γ-crystallin aggregation and investigated its mode of action.

We have shown that *myo*-inositol is a kinetic inhibitor of human γD-crystallin aggregation. It has no observable binding site on the native structure (*Figure 2A and B*) and only a very minor effect on protein stability (*Figure 2C and D*) or on the morphology of mature aggregates (*Figure 4*). Yet,

we found evidence that it suppresses an aggregation precursor or early aggregate (*Figure 2G and H*) and preferentially depletes small aggregates (*Figure 4A and D*). Further investigation in redox buffers pointed to disruption of the rate-limiting bimolecular interaction that initiates aggregation (*Figure 6D*). Disfavoring the bimolecular interaction is expected to indirectly favor reduction and refolding of the misfolded monomeric aggregation precursor that is kinetically trapped by non-native disulfide bonding – and this is what we observed (*Figure 6B and C*). While inositol was an effective aggregation suppressor even in fully oxidizing buffer (*Figure 1*), it synergized with the glutathione redox couple to suppress aggregation more strongly under mild-to-intermediate levels of oxidation (*Figure 5* and *Figure 6A*). This suggests especially effective kinetic inhibition of the earliest steps on the aggregation pathway of γ-crystallins during middle age, when the lens fiber cells' cytoplasm is only mildly oxidizing, to delay and slow the development of lens turbidity.

We refer to the proposed target of inositol as a rate-limiting bimolecular interaction, rather than as a dimeric aggregation nucleus, because we have previously shown that the aggregation process is misfolding-limited, not nucleation-limited (*Serebryany et al., 2016b*). (That is, 'seeding' the reaction by adding pre-aggregated protein to fresh protein does not make the latter aggregate more rapidly.) Why is the rate-limiting interaction bimolecular, even though the non-native disulfide bond that locks the aggregation precursor is intramolecular? We have previously carried out extensive atomistic modeling of the misfolding process, which showed detachment of the N-terminal β-hairpin from the W42Q variant and reannealing to non-native locations (*Serebryany et al., 2016b*). Yet, the intramolecular disulfide bond most favored for promoting aggregation could only form within the context of a dimer that reconstituted the topologically rearranged monomer in a manner akin to domain swapping (*Serebryany et al., 2016b*). (The Cys-Cys distance was too large in the rearranged monomer.) Thus, a dimeric interaction may be the catalyst for non-native disulfide bonding, especially in redox buffers, as indicated in Model 2 of *Figure 6C*. We have not yet obtained direct experimental evidence as to the nature of *myo*-inositol's interaction with this dimeric intermediate. Inositol may affect the bimolecular interaction in one of three ways: (1) inhibit formation of a dimer; (2) promote dissociation of the dimer; or (3) inhibit a key misfolding event that takes place within the context of the dimer. Determining which of these mechanisms predominates will likely require observing the key dimeric interaction more directly than we are currently able to do. In all three cases, however, inositol lowers the concentration of dimer that is productive for the aggregation process, and thereby either disfavors formation or favors reduction of the non-native intramolecular disulfide bonds (*Figure 6B*).

Another apparent paradox is seen in *Figure 7*: If inositol inhibits only the early stages of aggregation (i.e. the bimolecular interaction), why is it effective as an aggregation suppressor even after most of the turbidity has already developed? We have previously observed that mature coalesced aggregates settle to the bottom of the reaction vessel during the course of the experiment (*Serebryany and King, 2015*; *Serebryany et al., 2016a*) – an effect that also occurred in the current study but did not significantly affect turbidity measurements due to the vertically oriented optics. We hypothesize that globular aggregate particles coalesce and settle to the bottom of the plate or cuvette and largely cease growing thereafter. Then, continued rise in turbidity depends on continued formation and coalescence of new aggregate particles even as the soluble monomer is gradually depleted. Thus, the microscopic rate-limiting bimolecular interaction determines the number of particles that can finally precipitate as large light-scattering globules. This is consistent with *Figure 4D and F*, where the number of aggregate particles decreases in 100 mM inositol even though the size of the larger particles does not. The combination of misfolding-limited aggregation kinetics and precipitation of coalesced particles explains why inositol can still inhibit further turbidity development even in already-turbid solutions. If this mechanism translates to the lens, *myo*-inositol could be used to slow down cataract progression.

Additional research is needed to determine whether *myo*-inositol suppresses aggregation of other lens crystallins to a comparable degree and whether other compounds based on the inositol scaffold are even better aggregation suppressors. For example, since redox chemistry plays a large role in βγ-crystallin aggregation (*Serebryany et al., 2016b*; *Serebryany et al., 2018*), and glutathione fails to diffuse into the aging lens (*Lim et al., 2020*), replacing one hydroxyl of *myo*-inositol with a thiol might yield a bifunctional anti-cataract drug – an aggregation suppressor and a reducing agent. The inositol transporters expressed in the lens are quite selective, yet one of them (SMIT1) has been shown to import inositols modified with chlorine, fluorine, or azide in at least one position (*Fenili et al.,*

*2011*). Free inositol concentrations and inositol transport capacity need to be measured in healthy lenses across the human lifespan, as existing datasets have focused on aged cataractous lenses for the former and young lenses for the latter. The observation that young lenses have a lower free water content (*Heys et al., 2008*) suggests that effective [*myo*-inositol] is highest in young age.

A typical human diet is estimated to contain 1–2 g of *myo*-inositol per day, and ~4 g/day is produced endogenously from glucose in the liver and kidney (*Fisher et al., 2002*). Of all human tissues in which *myo*-inositol levels have been studied, the lens appears to have the highest, while the second-highest levels are found in the cytoplasm of neurons in the brain (*Fisher et al., 2002*). High brain inositol has been ascribed to the need for signaling processes via second-messenger cascades (*Fisher et al., 2002*), but it may also act as a chemical chaperone there. Thus, both *myo*- and *scyllo*-inositol, as well as some derivatives, have been shown to directly bind and stabilize oligomers of $A\beta_{42}$, inhibiting amyloid aggregation (*McLaurin et al., 2000*; *Sun et al., 2008*). *Scyllo*-inositol has been clinically studied in Alzheimer's disease in humans, with some promise as a disease-modifying agent (*Tariot et al., 2012*). Our finding of *myo*-inositol's likely aggregation-suppression role in the lens calls for more detailed investigation of whether such activity is general for β-sheet-rich proteins.

Several animal and human studies have observed a relationship between *myo*-inositol levels and cataract. Thus, in rats with streptozotocin- or galactose-induced cataracts, dietary supplementation with *myo*-inositol dramatically decreased the rate of cataract development: by 14 weeks of age, the untreated animals had totally opaque lenses, while the treated showed only initial stages of cataractogenesis (*Beyer-Mears et al., 1989*). In another study, dietary supplementation with *myo*-inositol triphosphate delayed cataract onset by ~44% in streptozotocin-induced diabetic rats (*Ruf et al., 1992*). The premise of the rat studies was that *myo*-inositol might act as an inhibitor of aldose reductase in the lens, but this was found not to be the case (*Beyer-Mears et al., 1989*). Direct inhibition of protein aggregation was not considered at the time. Dosage of *myo*-inositol is likely to be important in any potential cataract treatments, as genetic overexpression of its native lens transporter in mice resulted in cataract due to hyperosmotic stress (*Jiang et al., 2000*). Trisomy-21 (Down's syndrome), which leads to overexpression of the same transporter in humans, is likewise associated with premature cataract (*Chhetri, 2019*). Given that *myo*-inositol is severely depleted in cataractous human lenses but not in age-matched controls (*Tsentalovich et al., 2015*; *Yanshole et al., 2019*), and given that it is one of the very few molecules able to permeate into lens tissue, maintaining or restoring healthy levels of lens *myo*-inositol should be considered a promising strategy for delaying the onset of visual impairment or blindness due to cataract and thus reducing the global burden of this highly prevalent disease.

## Materials and methods

### Protein expression and purification

Wild-type and mutant human γD crystallin, without any tags, was overexpressed in BL21 RIL *E. coli* from a pET16b plasmid in SuperBroth medium (Teknova, Hollister, CA) and purified by ammonium sulfate fractionation, ion exchange, and size exclusion as described previously (*Serebryany et al., 2018*). For HSQC NMR, $^{15}$N-labeled proteins were produced by culturing the cells as above, then centrifuging and resuspending in M9 minimal medium containing 2 g/L unlabeled glucose and 1 g/L $^{15}$N-labeled ammonium acetate (Cambridge Isotope Labs, Tewksbury, MA) and induced with 1 mM IPTG. Due to leaky expression prior to the shift to labeled medium, this expression method resulted in abundant but partially labeled samples. Intact protein electrospray ionization MS showed ~60% $^{15}$N labeling in ~20% of the WT sample and ~70% $^{15}$N labeling in ~60% of the W42Q sample, with the balance entirely unlabeled. Concentrations for NMR experiments were chosen accordingly to obtain sufficient signal.

### Aggregation assays (turbidimetry)

Aggregation was induced by heating at 37 °C (unless otherwise indicated) in the presence of either 0.5 mM oxidized glutathione as described previously (*Serebryany et al., 2018*) or of redox buffer composed of reduced and oxidized glutathione (*Serebryany et al., 2016b*). Good's buffers were added to a final 10 mM concentration to set sample pH to a desired level: MES for pH 6.0, PIPES for pH 6.7, and HEPES for pH 7.4, with 150 mM NaCl in all cases. Experiments in the main text were

conducted in PIPES pH 6.7 buffer, unless otherwise indicated, because intralenticular pH is known to be slightly below neutral (**Bassnett and Duncan, 1988**). One mM EDTA was added to all aggregation reactions to scavenge any trace metals, since some transition metals can promote HγD aggregation (**Quintanar et al., 2016**). All compounds tested as aggregation inhibitors were of high purity. *Myo*-inositol (>99%), *scyllo*-inositol (>98%), glucose (>99%), sucrose (>99.5%), trehalose (>99%), and rhamnose (>99%) were purchased from MilliporeSigma (Burlington, MA). Galactose (98%) and arabinose (99%) were from Alfa Aesar (Heysham, UK); mannitol (99%) from BeanTown Chemical (Hudson, NH); glycerol (99.7%) from Avantor (Radnor, PA); and IPTG (99%) from Omega Scientific (Tarzana, CA).

To extract the maximum aggregation rate, linear tangents were fitted empirically to the turbidity traces (**Serebryany et al., 2019**), and the slope of the best-fit tangent to the steepest part of each curve was used as the maximum rate of aggregation. The x-intercept of that tangent was defined as the apparent lag time (see **Figure 6—figure supplement 2**; **Borgia et al., 2013**; **Serebryany et al., 2019**).

## Solution rheometry

To test whether 100 mM compounds (as in **Figure 1B**) affect bulk solution properties, we carried out rheometry using the Discovery HR20 instrument (Waters). Viscosity vs. shear rate was measured by flow sweep for buffer with or without select small carbohydrates at 100 mM concentration at 25 °C with temperature control on a 40 mm Peltier plate (parallel) with solvent trap. As positive control (to observe increase in viscosity) we used 20% glucose. Samples were loaded (1300 µl) with 2000 µm loading gap with rotations maintained at 1 rad/s to ensure proper distribution of the fluid/solution on the plate. Measurements were made with 1000 µm geometric gap with a prior trim gap offset of 50 µm. The expansion co-efficient was recorded as 0.0 µm/°C with a friction of 0.301432 µN·m/(rad/s) and geometric inertia of 8.86649 µN·m·s². The controlled axial force was maintained at 5.00000 N with flow torque limit of 0.0100000 µN·m, velocity limit of 1.00000 µrad/s and flow velocity tolerance of 5.00%.

## NMR data collection and analysis

NMR experiments were performed on a Bruker Avance II 600-MHz spectrometer equipped with a cryoprobe. HγD WT and W42Q samples were diluted to 45 µM and 30 µM $^{15}$N-labeled protein, respectively, in PBS pH 7, 0.1 mM EDTA, 10% D$_2$O, with or without 100 mM myo-inositol. DTT (2 mM) was added to γD W42Q to prevent aggregation. HSQC spectra were collected at 310 K. Spectra were processed using NMRPipe and data analysis was performed with NMRViewJ.

## Differential scanning fluorometry

Samples were buffer exchanged into 10 mM PIPES pH 6.7 buffer with no added salt (estimated 15 mM [Na$^+$] final) by gel filtration and used at 40 µM for the thermal melts. Lack of salt was found empirically to minimize aggregation. A reducing agent (1 mM tricarboxyethyl phosphine (TCEP)) was added to prevent oxidative misfolding. 1 X SYPRO Orange (Thermo Fisher) was added as the hydrophobicity probe. Melts were carried out using Bio-Rad CFX 96 Touch thermocyclers, with temperature ramping of 1 °C per minute between 25°C and 95 °C. Samples were prepared by mixing a 2 X protein sample with the appropriate ratio of aqueous myo-inositol and water and a constant amount of PIPES buffer to avoid any variation in pH or salinity among samples.

## Transmission electron microscopy

Samples were prepared in 10 mM PIPES pH 6.7 with 150 mM NaCl with 1 mM EDTA and 0.5 mM oxidized glutathione (except the control sample, which lacked glutathione) and incubated at 37 °C in the presence or absence of myo-inositol as indicated for 4 hr. They were then kept at room temperature for ~2 hr before being applied to carbon-coated 200-hex copper grids (EMS, Hatfield PA). Freshly ionized grids were floated on a 10 µl drop of sample for 1 min, then washed with 5 drops of 2% acidic uranyl acetate, and excess uranyl acetate was drawn off with grade 50 Whatman filter paper. Grids were allowed to dry and imaged with a Hitachi 7800 at 100KV. For maximal precision, aggregate classification and morphometric measurements were carried out manually using contour and area tracing in ImageJ on a stylus-equipped touchscreen. To minimize human bias during this morphometric analysis, a double-blind was implemented as described in Results.

## Thermal scanning Raman spectroscopy

Raman spectroscopy as a vibrational spectroscopy technique provides a host of structural information and has been a popular choice for spectroscopy of proteins. It is non-destructive, very sensitive to secondary

structure, and (unlike FT-IR) largely devoid of water in the crucial Amide I band of the Raman spectrum, where distinct secondary structures have distinct signatures (*Lippert et al., 1976*; *Rygula et al., 2013*; *Sane et al., 1999*). Additional information rich components like Tryptophan Fermi doublets and phenylalanine pockets make Raman a useful technique in probing tertiary structure, as well (*Wen, 2007*). However, thermal scanning with in-solution proteins is a challenging task not typically attempted with Raman due to sample evaporation and lack of precise temperature control. Thus, we developed a novel apparatus to serve as a thermal scanning Raman spectrometer, as described below.

Raman spectroscopy was carried out using a Horiba XploRa confocal Raman microscope, a detector thermoelectrically cooled to –70 °C, and a 1200 gr/mm grating blazed at 750 nm. A solid-state 785 nm laser was used for excitation, with an acquisition time of 180 s for all measurements and 1 cm$^{-1}$ spectral resolution. Typical power levels at the sample were ~41 mW. To allow for automatic removal of cosmic rays, two back-to-back spectra were collected. The Horiba denoise algorithm in standard mode was used to slightly smooth the spectra. The fluorescent baseline was fit and subtracted using a polynomial fit. As the excitation wavelength was distant from any protein absorption bands, photobleaching or heat induced sample deterioration was not expected. The spectrometer slit was set to 200 μm. The confocal aperture was set to 500 μm. The system was calibrated to the 520.7 cm$^{-1}$ silicon reference sample before every set of measurements. 20 μM protein in 10 mM PIPES buffer pH 6.7 and 150 mM NaCl was used for the measurements with a sample volume of 20 μl. Samples with inositol had 200 mM inositol. Since the Raman microscope was confocal, the measurements could be made through a sealed PCR tube with no sample handling. The PCR tube was laid on its side on a microscope slide with a slight tilt to keep the protein solution in the bottom of the tube. The Raman microscope was then focused into the middle of the protein solution. For thermal scanning, a temperature stage was constructed by taping a 10 kΩ 25 W resistor (Digikey MP825-10.0K-1%-ND) with Kapton tape to the microscope slide. A thermocouple was placed between the PCR tube and the resistor to monitor sample temperature. DC voltage was applied to the resistor to heat the sample; by slowly increasing the voltage and monitoring the thermocouple readout, the temperature could be increased in a controlled manner. The PCR tube was secured on top of the resistor with Kapton tape. Kapton tape was essential for the setup to be able to reach 65 °C. (See *Figure 2E* for schematic of thermal stage setup.) Thermal scans were done on the same sample heated in steps. Buffer blanks were carefully taken with buffer and buffer +inositol at every temperature with exactly the same setup and procedure; subtracting the matched blanks was critical for quality spectral readout. All spectra were normalized with respect to the sharp peak at 330 cm$^{-1}$. To ensure the quality of the protein specific signal, we recorded spectra of WT and compared that with PDB 1HK0 (*Basak et al., 2003*). A comparison of the secondary structure contents as retrieved from our Raman Amide I spectra deconvoluted with Labspec 6 software showed comparable contents with 1HK0. We carried out the same quality check for the isolated N-terminal domain and found its structure content similar to the N-terminal domain of the full-length protein in PDB 1HK0 (*Figure 2—figure supplement 1*).

## Free thiol counting

PEGylation gel shift assays were carried out essentially as described previously (*Serebryany et al., 2018*). A stock solution of PEG(5000)-maleimide (MilliporeSigma, Burlington, MA) at 100 mM was prepared in pure DMSO. Immediately prior to protein labeling, NuPAGE 4 x LDS loading buffer (BioRad), 0.4 M MES buffer pH 5.8, 60% glycerol, water, and the PEG-maleimide stock solution were mixed in the ratio 10:6:6:5:5, respectively. (MES buffer was needed to neutralize the Tris buffer in NuPAGE, thus minimizing non-specific reactions of maleimide; glycerol was needed to ensure labeled samples settled into the gel wells for SDS-PAGE). End-point samples from the glutathione redox-buffered protein aggregation reactions were added to this mixture in a 1:4 ratio and incubated at 50 °C for 1 hr, then used directly for SDS-PAGE. Markers were prepared by limited PEGylation of WT HγD, which contains six free Cys residues: two separate incubations were carried out as above but using only 6:1 and 3:1 molar excess of PEG-maleimide to HγD, respectively, then mixed together and frozen immediately after the 50 °C incubation.

## Acknowledgements

This work was supported by the National Institutes of Health, grant R01EY030444 to E I S; grants F32GM126651 and K99GM141459 to E S; and grant R01EY017370 to R E K We also acknowledge the

Center for Nanoscale Systems at Harvard University and the NSF National Nanoscience Infrastructure Network.

---

## Additional information

### Funding

| Funder | Grant reference number | Author |
|---|---|---|
| National Institutes of Health | R01EY030444 | Sourav Chowdhury<br>David C Thorn<br>Eugene I Shakhnovich |
| National Institutes of Health | F32GM126651 | Eugene Serebryany |
| National Institutes of Health | K99GM141459 | Eugene Serebryany |
| National Institutes of Health | R01EY017370 | Christopher N Woods<br>Rachel E Klevit |
| National Science Foundation | National Nanoscience Infrastructure Network | Nicki E Watson<br>Arthur A McClelland |

The funders had no role in study design, data collection and interpretation, or the decision to submit the work for publication.

### Author contributions

Eugene Serebryany, Conceptualization, Data curation, Formal analysis, Funding acquisition, Investigation, Methodology, Visualization, Writing - original draft, Writing - review and editing; Sourav Chowdhury, Data curation, Formal analysis, Investigation, Methodology, Visualization, Writing - original draft, Writing - review and editing; Christopher N Woods, Formal analysis, Investigation, Visualization, Writing - review and editing; David C Thorn, Data curation, Formal analysis, Investigation, Writing - review and editing; Nicki E Watson, Arthur A McClelland, Investigation, Methodology, Writing - review and editing; Rachel E Klevit, Formal analysis, Methodology, Supervision, Writing - review and editing; Eugene I Shakhnovich, Conceptualization, Formal analysis, Funding acquisition, Methodology, Supervision, Writing - review and editing

### Author ORCIDs

Eugene Serebryany http://orcid.org/0000-0003-1066-7143
David C Thorn http://orcid.org/0000-0002-7332-2292
Arthur A McClelland http://orcid.org/0000-0003-4798-5954
Rachel E Klevit http://orcid.org/0000-0002-3476-969X
Eugene I Shakhnovich http://orcid.org/0000-0002-4769-2265

### Decision letter and Author response

Decision letter https://doi.org/10.7554/eLife.76923.sa1
Author response https://doi.org/10.7554/eLife.76923.sa2

---

## Additional files

### Supplementary files
• Transparent reporting form

### Data availability
Numerical data and fits used to generate the plots and fits of Figure 1B,C,D,E have been uploaded as source data files. Raw numerical measurements behind the graphs presented in Figure 4 are included in Figure 4-source data 1. All raw TEM images and instrument files have been uploaded to the Dataverse repository and are available at: https://doi.org/10.7910/DVN/BVRS9M. Full unedited gels used to generate Figure 6 - figure supplement 1 were uploaded as source data. The numerical data

and fits used to generate Figure 6D were uploaded as a source data file. All other data are contained in the manuscript.

The following dataset was generated:

| Author(s) | Year | Dataset title | Dataset URL | Database and Identifier |
|---|---|---|---|---|
| Serebryany E | 2022 | Negative-stain TEM of human gamma-D crystallin W42Q variant aggregates +/- myo-inositol | https://doi.org/10.7910/DVN/BVRS9M | Harvard Dataverse, 10.7910/DVN/BVRS9M |

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
