## [Editor Report]

Cataract is one of the most prevalent protein aggregation disorders and the leading cause of vision loss worldwide. As the eye lens lacks cellular or protein turnover, efficient mechanisms have evolved to prevent protein aggregation for a lifetime. This study shows that an abundant metabolite of the eye lens, myo-inositol, plays an important role in preventing aggregation, acting as a chemical chaperone by an unusual mechanism.

---

## [Decision Letter]

**Decision letter after peer review:**

[Editors’ note: the authors submitted for reconsideration following the decision after peer review. What follows is the decision letter after the first round of review.]

Thank you for submitting your work entitled "A native chemical chaperone in the human eye lens" for consideration by *eLife*. Your article has been reviewed by 3 peer reviewers, and the evaluation has been overseen by a Reviewing Editor and a Senior Editor. The following individual involved in review of your submission has agreed to reveal their identity: Rachel E Klevit (Reviewer #2).

We are sorry to say that, after consultation with the reviewers, we have decided that your work will not be considered further for publication by *eLife*. However, the referees acknowledge the interest in your study and we would welcome a new submission, when, in due course, you have been able to address the various technical concerns. We anticipate that the necessary amount of additional work exceeds what can be expected in a normal revision for *eLife*.

Critical additional experiments include:

– More accurate Raman of FTIR spectra.

– *in vitro* aggregation assays based on amyloid-specific fluorescent dyes.

– Quantitative analysis of these aggregation assays.

*Reviewer #1 (Recommendations for the authors):*

This work reports the effect of myo-inositol on the aggregation of γ crystallin. Although in principle this is an important observation, the conclusions are speculative based on the evidence provided. More evidence should be provided on the overall effect of myo-inositol on γ crystallin aggregation, as well as on the mechanism by which this metabolite acts on the aggregation process.

The conclusion from the turbidity measurements in Figure 1 that the compounds investigated act by inhibiting the aggregation process, not by changing the aggregate morphology, should be confirmed. The TEM analysis in Figures 2 and 3 may not support this conclusion, since morphological differences are detected.

The quantification of the aggregate mass at the end of the reaction seems to be inconsistent from the turbidity and TEM analyses. A plot with the correlation of the two estimates as a function of the concentration of myo-inositol should be provided.

In Figure 1C the procedure to calculate the aggregation rate should be defined. The aggregation mechanism proposed in Figure 4 is still speculative based on the evidence reported.

The Raman spectra in Figure 6 are of low quality, in particular in panels G and H in the presence of myo-inositol. A quantitative analysis of the secondary structure should consider also the second derivative of the spectra. The conclusions about the conformational transitions and the mechanism of aggregation are therefore not supported by the data.

*Reviewer #2 (Recommendations for the authors):*

Results are presented that establish the ability of myo-inositol to suppress aggregation of γD-crystallin *in vitro* at concentrations found in lens. Having demonstrated this point, the remainder of the study attempts to characterize the effects on aggregate structure (by negative stain electron microscopy) and at a more molecular level (by differential scanning fluorimetry and Raman spectroscopy).

The strengths of the manuscript lie in the intriguing hypothesis being tested and the data presented in Figures 1 and 2. The notion that an abundant metabolite could serve as a chemical chaperone in the special environment of eye lens is interesting and the *in vitro* data presented in Figure 1 are consistent with the proposal. The effects of differing concentrations of myo-inositol on the nature of aggregates ("globular" versus "fibrillar") are confusing and the attempts to explain the observations are "hand-waving" at best. It does not appear that the species dubbed "fibrils" have been verified to be fibrils according to the standard definition and the EM images do not, by eye, appear fibrillar. This aspect of the study does little to shed light on the process under investigation. The authors attempt to ascertain whether the observed effects of myo-inositol are direct and, if so, how the metabolite affects crystallin. The approaches used, DSF and Raman, do not provide much insight. The effects detected by DSF (Figure 5) are very small and difficult to interpret. Without more context, it is impossible to know whether the observations are specific to γD-crystallin, or whether the metabolite at concentrations up to 200 mM similarly affect melting transitions of other proteins. The Raman spectroscopy is also difficult to judge. The band at 1650 cm-1 is broad and therefore difficult to measure intensity for. The authors should consider using other types of spectroscopy (CD, NMR for example) to characterize the effects of myo-inositol on γD-crystallin. Without more robust and compelling evidence for an interaction, the paper is reduced to one that reports intriguing observations with little or no mechanistic insights.

1. If the assignment of species of aggregates as fibrillar is an essential part of the model, then data such as ThT fluorescence and/or other fibril-specific parameters are needed. The current method of handling the images seems rather ad hoc.

2. The DSF data are not convincing. The effects are small and how the presence of the fluorescent dye molecule alters the system is an unknown. Wouldn't it be more informative to compare WT- γD-crystallin (which is not aggregation-prone on a laboratory timescale) to mutant forms? As well, inclusion of a similar comparison but with circular dichroism should be seriously considered.

3. γD-crystallin is amenable to solution-state NMR, providing the opportunity to obtain residue-level information on the direct effects of myo-inositol. There are (at least) two different types of NMR experiments that would be informative. First are standard 1H, 15N-HSQC spectra as a function of increasing metabolite, compared with a similar series where a non-effective/less-effective reagent (Figure 1B) is added. A second NMR-based approach would be time-dependent 1D-NMR aggregation assays similar to those used in (Baughman et al. (2018) JBC 293:2687). These could allow the authors to determine whether the effects of myo-inositol occur early on the aggregation pathway, as they suggest in their model (but currently do not have experimental evidence for).

*Reviewer #3 (Recommendations for the authors):*

Serebryany et al. here investigated whether myo-inositol could be a physiological chemical chaperone of lens crystallins and thus whether myo-inositol or a derivative thereof could be considered as a drug candidate for the prevention of cataract. This investigation is inspired by the observation that myo-inositol levels are high in healthy lens epithelial cells and are substantially lower in cataractic lenses.

Myo-inositol is generally regarded as an osmolyte and is also a secondary messenger of the insulin pathway and is therefore seen to drop in diabetes in all insulin-sensitive cell types. The hypothesis considered here should therefore be able to convincingly demonstrate that myo-inositol is a bona fide chemical chaperone and not an osmolyte.

Chemical chaperones are often small molecule active site analogs that bind specifically and with a reasonably high affinity (low μm to nM) to their target so that they can sufficiently stabilize the native state. The authors argue that myo-inositol is a chemical chaperone because (1) "aggregation suppression is already significant in the low-mM range, when [myo-inositol] is well below 1% w/v (Figure 1A)". Experiments are performed with mM inositol concentrations and μm protein concentrations. In such molar excess observed inhibition of aggregation can still simply be due to osmotic effects (2) "compositionally similar compounds have highly distinct effects; thus, myo-inositol (C6H12O6) is a much stronger suppressor than mannitol (C6H14O6), while galactose and IPTG have opposite effects (Figure 1B)." Again, at such molar excess this does not demonstrate specificity but could equally be due to differences in osmotic effects.

Finally, the TEM images, DSF or Raman spectroscopy data also do not support chemical chaperoning to the exclusion of osmotic stabilization. Both TEM and DSF provide "paradoxical" results and the secondary structural consolidation of a poorly structured part of native crystallin by myo-inositol seen by Raman spectroscopy can again be explained by excluded volume effects by osmosis.

Overall, while mM excess concentrations of myo-inositol certainly contribute to the reduction of g-crystallin aggregation *in vitro* (and probably in vivo) conclusive evidence for chemical chaperoning by myo-inositol is not provided.

To convince readers that myo-inositol is a chemical chaperone the authors should be able to demonstrate direct binding by identifying the binding site on the protein by structural characterization, quantifying direct binding (ITC, SPR, thermophoresis,…) and demonstrate the SAR analysis discussed in Figure 1B is comprehensible on the basis of that information.

[Editors’ note: further revisions were suggested prior to acceptance, as described below.]

Thank you for submitting your article "A native chemical chaperone in the human eye lens" for consideration by *eLife*. Your article has been reviewed by 2 peer reviewers, and the evaluation has been overseen by a Reviewing Editor and David Ron as the Senior Editor. The reviewers have opted to remain anonymous.

Essential revisions:

The reviewers agree that the revised paper is substantially improved and will become suitable for publication in *eLife* with some additional revisions:

1. The conclusion about the mechanism of action of myo-inositol remains speculative at this stage. Specifically, the nature of the interaction of myo-inositol with the intermediate dimeric species remains unclear. This limitation should be stated clearly in the final manuscript. Additional experiments to address this point are considered beyond the scope of this study.

2. In addition, please address the following points:

– The solubility of myo-inositol is reported as 500 mM. It is not clear, however, whether self-association into oligomers could be present at lower concentrations. This is important since the effect of myo-inositol may be due to this self-association.

– A quantitative plot of the chemical shift changes should be presented in Figure 2. From the spectra shown it is not necessarily evident that the results are incompatible with binding.

– The curves in Figure 7S1 are described as reproducibly multiphasic. However, it is not clear how the reproducibility was assessed.

*Reviewer #1 (Recommendations for the authors):*

The authors investigated whether a metabolite could be responsible for the inhibition of the aggregation of gammaD-crystallin in the lens. They found that myo-inositol can act to achieve this effect.

They report data that support the conclusion that myo-inositol does not bind the native and aggregate states of gammaD-crystalling, thus suggesting that myo-inositol may be characterised as a chemical chaperone.

A weakness of this work is the proposed mechanism of action of myo-inositol. The authors conclude that myo-inositol inhibit the formation of a dimeric intermediate in the aggregation pathway. However, this mechanism of action may be considered still speculative.

The identification of a metabolite capable of inhibiting the aggregation of a protein involved in cataract formation may create novel therapeutic opportunities.

The authors should be commended for having significantly improved the manuscript with respect to the previous version.

However, the conclusion about the mechanism of action of myo-inositol seems speculative at this stage. The nature of the interaction of myo-inositol with the intermediate dimeric species remains unclear. In alternative to binding studies, which are highly challenging for intermediate states, the authors could devise a way to assess the effect of myo-inositol on the rate constant of dimerization. Otherwise, other kinetic mechanisms may be possible.

*Reviewer #2:*

This is a vastly improved version of the original manuscript with significant additional and detailed data excluding native, unfolded state or aggregated state remodelling but indicating that myo-inositol functions as a chemical chaperone affecting a rate-limiting biomolecular step on the aggregation pathway . The authors have convincingly addressed my concerns.

[Editors’ note: further revisions were suggested prior to acceptance, as described below.]

Thank you for resubmitting your work entitled "A native chemical chaperone in the human eye lens" for further consideration by *eLife*. Your revised article has been evaluated by David Ron (Senior Editor) and a Reviewing Editor.

The manuscript has been improved but there is one remaining issue that should be addressed prior to acceptance, as outlined below:

*Reviewer #1 (Recommendations for the authors):*

The authors have revised the manuscript in a satisfactory manner concerning the evidence that they have about the mechanism of action.

However, statement in the abstract is not yet fully consistent with the evidence provided:

"Unlike many known chemical chaperones, myo-inositol's primary target was neither the native nor the unfolded state of the protein, nor the final aggregated state, but rather the rate-limiting bimolecular step on the aggregation pathway."

The authors should revise this statement to reflect that this is not a fact, but as a possible interpretation of their results.

---

## [Author Response]

[Editors’ note: the authors resubmitted a revised version of the paper for consideration. What follows is the authors’ response to the first round of review.]

Critical additional experiments include:– More accurate Raman of FTIR spectra.

Much higher-resolution Raman spectra were obtained. We determined that the initial poor resolution resulted from buffer subtraction artifacts. A full thermal ramp of plain buffer with and without inositol under the same conditions as the protein measurements was used for background subtraction and has greatly improved the spectral quality. The new spectra are now shown in the new Figure 2F,G,H.

– *in vitro* aggregation assays based on amyloid-specific fluorescent dyes.

This and related critiques by the individual Reviewers stem from an unfortunate confusion of terms, for which we apologize. The aggregation of human γD-crystallin in our study is not amyloid, as we have shown in a series of previous papers starting in 2015, including the finding that these aggregates do not stain with the amyloid specific dye Thioflavin T. We have removed all descriptions of the aggregates as “fibrillar” to avoid confusing any readers due to the frequent conflation of “fibrils” with “amyloids” in common scientific usage. Since the aggregates are not amyloid, the approaches traditionally used for characterizing amyloids do not apply to our system. By contrast, solution turbidity is uniquely well-suited to characterize this system because it is (1) quantitative, as we have shown in prior work and further confirmed in this work, and (2) the most physiologically relevant observable, since cataract is lens turbidity. We have added extensive discussion of the nature and physiological relevance of these aggregates to the Introduction section. We realize that in the minds of many readers protein aggregation is now nearly synonymous with amyloid formation, and we welcome the opportunity to draw attention to an aggregation process that is well-characterized and operates by a very different set of principles.

– Quantitative analysis of these aggregation assays.

As we have noted above, the aggregates are not amyloid in nature. However, our existing solution turbidity assays provide a very strong basis for quantitative analysis because solution turbidity is a very strong correlate of total aggregated mass of the protein in our experimental system. We have extensively revised both the Introduction and the Discussion to refer to the relevant prior work while clearing up any misunderstanding of the nature of the aggregation process. We have also added more quantitative data on the turbidity measurements themselves, such as maximum rates and lag times with and without inositol. These are now included in the revised Figure 1, as well as the all-new Figures 5 and 6. Other techniques used in our study (negative-stain TEM, Raman spectroscopy, and thiol counting) provide additional qualitative or quantitative insights into various aspects of the aggregation process, leading to a new and surprising mechanistic model proposed in Figure 6.

Reviewer #1 (Recommendations for the authors):This work reports the effect of myo-inositol on the aggregation of γ crystallin. Although in principle this is an important observation, the conclusions are speculative based on the evidence provided. More evidence should be provided on the overall effect of myo-inositol on γ crystallin aggregation, as well as on the mechanism by which this metabolite acts on the aggregation process.

We thank the Reviewer for recognizing the importance of the motivating observation, and we have included much additional data on the mode of action of inositol. In particular, we now know that this compound has little effect on native-state thermodynamic stability – in contrast to the most established theoretical models of osmolyte effects. Nor does it affect the morphology of the aggregates or their size distribution enough to explain the aggregation suppression. Instead, we present evidence that it specifically targets the rate-limiting non-native bimolecular interaction on the aggregation pathway and thereby indirectly shifts the distribution of redox states among the protein molecules, disfavoring the non-native internal disulfide bonds required for aggregation.

The conclusion from the turbidity measurements in Figure 1 that the compounds investigated act by inhibiting the aggregation process, not by changing the aggregate morphology, should be confirmed. The TEM analysis in Figures 2 and 3 may not support this conclusion, since morphological differences are detected.

We have confirmed this observation in several ways. First, the strong linear inverse correlation of solution turbidity and total remaining mass of soluble protein (Figure 1B) would have been very unlikely if either the shape or the size distribution of the aggregates were greatly affected. Second, the newly obtained high-quality Raman spectra in Figure 2 clearly show non-native secondary structure formation during the thermal ramp without inositol, and near-total suppression thereof by inositol. Third, we have noted that the morphological differences seen by TEM are very minor and occur predominantly at the very low end of aggregate particle sizes. Light-scattering (solution turbidity) is dominated by the large particles, among which very little difference is observed between 0 and 100 mM inositol.

The quantification of the aggregate mass at the end of the reaction seems to be inconsistent from the turbidity and TEM analyses. A plot with the correlation of the two estimates as a function of the concentration of myo-inositol should be provided.

We believe the turbidity data are much more quantitative than the TEM data in this regard. They are certainly not contradictory: a visual comparison of the TEM grid surveys in Figure 3B (no inositol) and Figure 3C (100 mM inositol) shows less extensive aggregation in the latter condition. This is less clear from the distributions of aggregate particle sizes in Figure 4. The likely explanation is as follows. Close-up images were needed for the morphometry, yet negative-stain TEM is notorious for the possibility of human bias in which specific spots on the grid get selected for high-magnification imaging, even when the total number of images is high (103 in our study). Therefore, we set up a double-blind: the microscopist (N. E. Watson) was given eight samples numbered in random order – two each from 0, 100, and 250 mM inositol, plus two controls, and instructed to take high magnification images as representative of each sample as possible. The images themselves were then stripped of sample numbers and randomized (by D. C. Thorn) before morphometry was carried out (by E. Serebryany). Naturally, the human eye gravitates toward imaging spots that are not empty, and such spots also ensure proper focusing of the optics, so it is entirely plausible that high-magnification images that contained aggregates were over-represented to some degree for samples containing 100 mM inositol. For this reason, statistical comparison of the ranked particle sizes (by the K-S test) was only carried out with respect to the shapes of the distributions themselves. We consider these double-blinded high-magnification TEM images a good technique for quantifying aggregate morphology but not a reliable measure of the overall extent of aggregation.

In Figure 1C the procedure to calculate the aggregation rate should be defined. The aggregation mechanism proposed in Figure 4 is still speculative based on the evidence reported.

We have added a definition of the calculation to the relevant Methods section, and we have also included a graph of one fit as an example (panel E of the new Figure 6—supplement 2). We acknowledge that the proposed macroscopic mechanism of aggregation (from misfolded precursors, to small extended aggregates, to collapse and coalescence, and finally precipitation) is somewhat speculative, since it is based on end-point snapshots by TEM. We believe it is the most likely mechanism given the data. However, we have modified this scheme (which is now Figure 3H) to separate the proposed model of the macroscopic aggregation process from the microscopic interactions that are the primary target of inositol. We have presented two alternative (though related) models for the latter process in the new Figure 6C.

The Raman spectra in Figure 6 are of low quality, in particular in panels G and H in the presence of myo-inositol. A quantitative analysis of the secondary structure should consider also the second derivative of the spectra. The conclusions about the conformational transitions and the mechanism of aggregation are therefore not supported by the data.

We agree with the Reviewer that the quality of the initial Raman spectra was not optimal. Thermal-scanning Raman spectroscopy is not simple and required a custom home-built apparatus, whose properties took time to understand. However, we now have much higher-quality Raman spectra, shown in the new Figure 2, thanks to the changes described in response to the editorial summary above.

The improved Raman spectra also revealed that, indeed, the mechanism of conformational stabilization that we had proposed on the basis of those initial spectra was not correct. We now can clearly resolve the non-native βstructures that form during the course of the thermal ramp in the absence of inositol and are suppressed by inositol, and we have revised the model and discussion accordingly.

Reviewer #2 (Recommendations for the authors):Results are presented that establish the ability of myo-inositol to suppress aggregation of γD-crystallin *in vitro* at concentrations found in lens. Having demonstrated this point, the remainder of the study attempts to characterize the effects on aggregate structure (by negative stain electron microscopy) and at a more molecular level (by differential scanning fluorimetry and Raman spectroscopy).The strengths of the manuscript lie in the intriguing hypothesis being tested and the data presented in Figures 1 and 2. The notion that an abundant metabolite could serve as a chemical chaperone in the special environment of eye lens is interesting and the *in vitro* data presented in Figure 1 are consistent with the proposal. The effects of differing concentrations of myo-inositol on the nature of aggregates ("globular" versus "fibrillar") are confusing and the attempts to explain the observations are "hand-waving" at best. It does not appear that the species dubbed "fibrils" have been verified to be fibrils according to the standard definition and the EM images do not, by eye, appear fibrillar. This aspect of the study does little to shed light on the process under investigation. The authors attempt to ascertain whether the observed effects of myo-inositol are direct and, if so, how the metabolite affects crystallin. The approaches used, DSF and Raman, do not provide much insight. The effects detected by DSF (Figure 5) are very small and difficult to interpret. Without more context, it is impossible to know whether the observations are specific to γD-crystallin, or whether the metabolite at concentrations up to 200 mM similarly affect melting transitions of other proteins. The Raman spectroscopy is also difficult to judge. The band at 1650 cm-1 is broad and therefore difficult to measure intensity for. The authors should consider using other types of spectroscopy (CD, NMR for example) to characterize the effects of myo-inositol on γD-crystallin. Without more robust and compelling evidence for an interaction, the paper is reduced to one that reports intriguing observations with little or no mechanistic insights.

We thank the Reviewer for the thoughtful critiques and have revised the paper in collaboration with this Reviewer (who is now a co-author) to address them, as described below.

1. If the assignment of species of aggregates as fibrillar is an essential part of the model, then data such as ThT fluorescence and/or other fibril-specific parameters are needed. The current method of handling the images seems rather ad hoc.

As noted above in response to the Editorial Summary, this aggregation process is not amyloid and does not elicit ThT fluorescence. We have removed all descriptions of the aggregates as fibrillar to avoid this confusion. We now describe the small aggregates as “extended.”

2. The DSF data are not convincing. The effects are small and how the presence of the fluorescent dye molecule alters the system is an unknown. Wouldn't it be more informative to compare WT- γD-crystallin (which is not aggregation-prone on a laboratory timescale) to mutant forms? As well, inclusion of a similar comparison but with circular dichroism should be seriously considered.

We agree that the effects observed by DSF were too minor to explain the effect of *myo-*inositol and that, in fact, the small magnitude of these effects points away from a mode of action via stabilization of the native state. We have thoroughly revised the interpretation of our results accordingly and introduced a much more relevant measure of misfolding (distributions of the number of free thiols per molecule) in the new Figure 6. We attempted label-free characterization of stability by differential scanning calorimetry, but the resulting traces were not interpretable due to an apparent phase transition – possibly precipitation of inositol at very high temperatures (data not shown).

3. γD-crystallin is amenable to solution-state NMR, providing the opportunity to obtain residue-level information on the direct effects of myo-inositol. There are (at least) two different types of NMR experiments that would be informative. First are standard 1H, 15N-HSQC spectra as a function of increasing metabolite, compared with a similar series where a non-effective/less-effective reagent (Figure 1B) is added. A second NMR-based approach would be time-dependent 1D-NMR aggregation assays similar to those used in (Baughman et al. (2018) JBC 293:2687). These could allow the authors to determine whether the effects of myo-inositol occur early on the aggregation pathway, as they suggest in their model (but currently do not have experimental evidence for).

We collaborated with the Reviewer to do exactly this. As suggested by the Reviewer, we expressed and purified ^15^N-labeled WT and mutant proteins, whose solution spectra at 37 °C with and without inositol were then obtained in the Reviewer’s lab. The spectra with and without inositol were fully superimposable for both the WT and the mutant, which provides gold-standard evidence that the compound does not affect the native state of either protein. This observation was crucial in helping us rule out native-state effects and properly focus instead on the misfolded aggregation precursor and the rate-limiting bimolecular interaction.

Reviewer #3 (Recommendations for the authors):Serebryany et al. here investigated whether myo-inositol could be a physiological chemical chaperone of lens crystallins and thus whether myo-inositol or a derivative thereof could be considered as a drug candidate for the prevention of cataract. This investigation is inspired by the observation that myo-inositol levels are high in healthy lens epithelial cells and are substantially lower in cataractic lenses.Myo-inositol is generally regarded as an osmolyte and is also a secondary messenger of the insulin pathway and is therefore seen to drop in diabetes in all insulin-sensitive cell types. The hypothesis considered here should therefore be able to convincingly demonstrate that myo-inositol is a bona fide chemical chaperone and not an osmolyte.

We thank the Reviewer for pointing out the broader importance of *myo-*inositol. The terms “osmolytes” and “chemical chaperones” have been used somewhat loosely. “Pharmacological chaperones” have sometimes been used to describe higher-specificity, higher-affinity interactions. We have used “chemical chaperone” in preference to “osmolyte” simply because the preponderance of available evidence points away from an osmotic effect, as discussed below. We have added a total of 33 references compared to the initial version of the article, most of which cite the relevant literature on osmolytes, chemical chaperones, and their mechanisms of action.

Chemical chaperones are often small molecule active site analogs that bind specifically and with a reasonably high affinity (low μm to nM) to their target so that they can sufficiently stabilize the native state. The authors argue that myo-inositol is a chemical chaperone because (1) "aggregation suppression is already significant in the low-mM range, when [myo-inositol] is well below 1% w/v (Figure 1A)". Experiments are performed with mM inositol concentrations and μm protein concentrations. In such molar excess observed inhibition of aggregation can still simply be due to osmotic effects (2) "compositionally similar compounds have highly distinct effects; thus, myo-inositol (C6H12O6) is a much stronger suppressor than mannitol (C6H14O6), while galactose and IPTG have opposite effects (Figure 1B)." Again, at such molar excess this does not demonstrate specificity but could equally be due to differences in osmotic effects.

In our reading of the literature, micro- and nanomolar affinity compounds have typically been termed “pharmacological chaperones,” but this may be a matter of preferred terminology. Crowders are thought to stabilize proteins via excluded volume effects, as the Reviewer notes, but concentrations on the order of 10 g/L are simply too low to cause any appreciable excluded volume effects (see, e.g., Christiansen and Wittung Schafshede, Biophys. J. 2013; Mukherjee et al., J. Phys. Chem. B 2015). The Reviewer may be referring instead to models of osmolyte effect via preferential exclusion from protein surfaces (as in Street, Bolen, and Rose, PNAS 2006). In either case, the net effect is thermodynamic stabilization of the native state. In our study, we observe at best very minor stabilization of the native state by DSF (Figure 2C, D), and we have observed the same by chemical denaturation with intrinsic Trp fluorescence detection (data not shown). Instead, the aggregation suppression is attributable to inhibition of the rate-limiting bimolecular interaction that stabilizes the misfolded conformer locked by a non-native disulfide bond (Figure 6). Finally, we have confirmed experimentally that 100 mM inositol is not sufficient to change bulk solvent properties, as shown in the new Figure 1—supplement 1.

Finally, the TEM images, DSF or Raman spectroscopy data also do not support chemical chaperoning to the exclusion of osmotic stabilization. Both TEM and DSF provide "paradoxical" results and the secondary structural consolidation of a poorly structured part of native crystallin by myo-inositol seen by Raman spectroscopy can again be explained by excluded volume effects by osmosis.

We have resolved the “paradoxical” results of Raman, as described above, by obtaining better-quality spectra. The new Raman data – and, moreover, the new NMR data – make it clear that inositol does not affect the structure of the native state. Our previous finding that it stabilizes the N-terminal domain was erroneous: we now know that aggregation of the protein during the course of the thermal ramp is what decreases the signal intensity at 1650 cm^-1^ more quickly without inositol. The new Raman data indicate than even “high” [inositol] in our study (200 mM) does not prevent the protein from unfolding, as seen in the smooth decline of total Raman signal intensity during the course of the thermal ramp without any new peaks emerging (Figure 2H). What is absolutely clear by comparison of the Raman datasets is that inositol suppresses conformational transitions to populate non-native secondary structures – first a peak at ~1630 cm^-1^ and subsequently the peak at ~1620 cm^-1^ that is typically associated with β-sheet-rich aggregated structures. Chemical chaperones can have a variety of mechanisms, including acting as inhibitors of nucleation or other early aggregation steps. We have added a new paragraph 1 to the Discussion to cite this.

Overall, while mM excess concentrations of myo-inositol certainly contribute to the reduction of g-crystallin aggregation *in vitro* (and probably in vivo) conclusive evidence for chemical chaperoning by myo-inositol is not provided.

We are open to clarifying terminology if the Reviewer believes the term “chemical chaperone” is not the most suitable description of the observed effects. We simply do not wish to mislead readers into thinking the observed effects are due to what is typically observed with osmolytes: stabilization of the native state.

To convince readers that myo-inositol is a chemical chaperone the authors should be able to demonstrate direct binding by identifying the binding site on the protein by structural characterization, quantifying direct binding (ITC, SPR, thermophoresis,…) and demonstrate the SAR analysis discussed in Figure 1B is comprehensible on the basis of that information.

As noted above, we have obtained HSQC NMR spectra demonstrating no interaction between inositol and the native state. Some chemical chaperones, however, are known to inhibit downstream processes such as nucleation of aggregates, and these are more relevant comparisons to our present findings.

[Editors’ note: what follows is the authors’ response to the second round of review.]

Essential revisions:The reviewers agree that the revised paper is substantially improved and will become suitable for publication in eLife with some additional revisions:1. The conclusion about the mechanism of action of myo-inositol remains speculative at this stage. Specifically, the nature of the interaction of myo-inositol with the intermediate dimeric species remains unclear. This limitation should be stated clearly in the final manuscript. Additional experiments to address this point are considered beyond the scope of this study.

We agree with the Reviewers about the need to state forthrightly this limitation of our study, and in fact we stated in paragraph 4 of Discussion that multiple mechanisms remain possible:

“Inositol may affect the bimolecular interaction in one of three ways: (1) inhibit formation of a dimer; (2) promote dissociation of the dimer; or (3) inhibit a key misfolding event that takes place within the context of the dimer. Determining which of these mechanisms predominates will likely require observing the key dimeric interaction more directly than we are currently able to do.”

To avoid confusion and further emphasize the Reviewers’ caveat, we have now prefaced this statement with the following sentence:

“We have not yet obtained direct experimental evidence as to the nature of *myo-*inositol’s interaction with this dimeric intermediate.”

2. In addition, please address the following points:– The solubility of myo-inositol is reported as 500 mM. It is not clear, however, whether self-association into oligomers could be present at lower concentrations. This is important since the effect of myo-inositol may be due to this self-association.

We thank the Reviewer for raising this point. Crystallization of *myo-*inositol is driven by hydrogen bonding,^1^ so if inositol oligomers (crystal seeds) do form as the concentrations in our experiments get closer to the ~500 mM solubility limit, we should expect formation of inositol-inositol hydrogen bonds.

Therefore, we carried out Raman spectroscopy using an excitation wavelength of 532 nm (at room temperature) on aqueous *myo-*inositol solutions ranging from 100 mM to 500 mM to investigate hydrogen bonding, both among inositol molecules and between inositol and water. Since the high-wavenumber region of O-H stretch for alcohols overlaps with, and is dominated by, the water O-H stretch, we focused on the 300 to 1100 cm^-1^ region, which includes C-O and C-C-O vibrational modes.^2^

First, to determine which Raman peaks of *myo-*inositol are sensitive to hydrogen bonding, we obtained spectra of *myo-*inositol dissolved in water and in D_2_O (Figure 1—supplement 3A). Peaks at 891 and 1071 cm^-1^ for *myo-*inositol in water shifted by -45 and -39 cm^-1^, respectively, in D_2_O. Peaks at 423 and 504 cm^-1^ each showed a minor shift (-8 cm^-1^) in D_2_O. Notably, the shifts were clear-cut, suggesting absence of OH-OH hydrogen bonds in D_2_O *myo-*inositol solutions.

Second, to probe for any emergent signature of inositol-inositol hydrogen bonds, we obtained Raman spectra of aqueous *myo-*inositol at five concentrations, from 100 to 500 mM (Figure 1—supplement 3B). Aside from concomitant increases in signal intensity for all peaks in the 300 to 1100 cm^-1^ region, no differences were observed among these spectra. To summarize, Raman spectroscopy revealed signatures of the expected inositol-water hydrogen bonds but none for direct inositol-inositol interactions in (or even above) the concentration range used in our crystallin aggregation experiments. However, we cannot rule out indirect interactions, like the probable water-mediated clusters in water-ethanol solutions.^3^

We have added Figure 1—supplement 3 and included the following sentences in the main text of the manuscript:

“In crystals, *myo-*inositol self-associates via networks of hydrogen bonds [ref: Rabinowitz and Kraut, 1964]. Although the maximum *myo-*inositol concentrations used in this study were not far below the empirical solubility limit of ~500 mM, Raman spectroscopy (Figure 1—supplement 3) did not reveal any signature of direct self-association of inositol in the 100 to 500 mM concentration range. However, as with any alcohol, transient water-mediated clusters [ref: Dolenko *et al.*, 2015] cannot be ruled out for inositol.”

– A quantitative plot of the chemical shift changes should be presented in Figure 2. From the spectra shown it is not necessarily evident that the results are incompatible with binding.

We have included a new Figure 2 – supplement 1 showing the distribution of chemical shift changes, as well as a new reference.

First, there is a systematic slight shift across all the peaks in the NMR spectra in the presence of inositol. This does not indicate a specific binding interaction because addition of any osmolyte in this concentration range is expected to produce a minor overall shift in the NMR spectrum due to the change in the solvent environment.^4^ We have added this note and reference to the text.

Second, to address the Reviewers’ request for a more quantitative plot of chemical shift changes, we have produced the following histograms, with are now included as Figure 2—supplement 1. All peak shifts are very small: <0.02 ppm, with a mean shift <0.01 ppm, for both W42Q and WT. We tested whether these additional very small shifts were randomly distributed. The Shapiro-Wilk test could not rule out a normal distribution for the CSPs of W42Q (*p = 0.13*) and indicated only marginal significance for WT (*p = 0.011*). We conclude that lack of a specific binding site remains the most likely interpretation.

– The curves in Figure 7S1 are described as reproducibly multiphasic. However, it is not clear how the reproducibility was assessed.

As stated in the legend to Figure 7—supplement 1, “Four technical replicates were carried out in parallel for each set of curves, and the curves shown consist of averages ± S.E.M. of the replicates.” In other words, reproducibility was assessed by comparing the aggregation curves from four separate reactions that were run in separate wells of the same 96-well plate at the same time. This observation would have been unexpected if prion-like (nucleation-driven) aggregation were occurring, since in that case formation of an initial nucleus that enables subsequent secondary nucleation tends to be highly stochastic. The small size of the error bars in the figure makes them difficult to see; we therefore revised the figure to add a zoom-in inset showing the error bars more clearly.

References

1. Rabinowitz, I. N. and Kraut, J. The Crystal Structure of Myo-Inositol. *Acta Cryst.* 17, 159-168 (1964).

2. Larkin, P. J. Infrared and raman spectroscopy: principles and spectral interpretation. (Elsevier, 2011).

3. Dolenko, T. A. *et al.* Raman Spectroscopy of Water-Ethanol Solutions: The Estimation of Hydrogen Bonding Energy and the Appearance of Clathrate-like Structures in Solutions. *Journal of Physical Chemistry A* 119, 10806-10815, doi:10.1021/acs.jpca.5b06678 (2015).

4. Iwaya, N. *et al.* Principal component analysis of data from NMR titration experiment of uniformly N-15 labeled amyloid β (1-42) peptide with osmolytes and phenolic compounds. *Archives of Biochemistry and Biophysics* 690, doi:10.1016/j.abb.2020.108446 (2020).

[Editors’ note: what follows is the authors’ response to the second round of review.]

Reviewer #1 (Recommendations for the authors):The authors have revised the manuscript in a satisfactory manner concerning the evidence that they have about the mechanism of action.However, statement in the abstract is not yet fully consistent with the evidence provided:"Unlike many known chemical chaperones, myo-inositol's primary target was neither the native nor the unfolded state of the protein, nor the final aggregated state, but rather the rate-limiting bimolecular step on the aggregation pathway."The authors should revise this statement to reflect that this is not a fact, but as a possible interpretation of their results.

We apologize for the oversight and have corrected the sentence in the abstract to the following:

“Unlike many known chemical chaperones, myo-inositol's primary target was not the native, unfolded, or final aggregated states of the protein; rather, we propose that it was the rate-limiting bimolecular step on the aggregation pathway.”